# SEC14-like condensate phase transitions at plasma membranes regulate root growth in Arabidopsis

Chen Liu[1,2,3], Andriani Mentzelopoulou[1,2], Fotini Papagavriil[1,2],
Prashanth Ramachandran[4], Artemis Perraki[2], Lucas Claus[5,6], Sebastian Barg[7],
Peter Dörmann[8], Yvon Jaillais[9], Philipp Johnen[10], Eugenia Russinova[5,6], Electra Gizeli[1,2],
Gabriel Schaaf[10], Panagiotis Nikolaou Moschou [1,2,3]*

1 Department of Biology, University of Crete, Heraklion, Greece, 2 Institute of Molecular Biology and
Biotechnology, Foundation for Research and Technology-Hellas, Heraklion, Greece, 3 Department of Plant
Biology, Uppsala BioCenter, Swedish University of Agricultural Sciences and Linnean Center for Plant
Biology, Uppsala, Sweden, 4 Department of Organismal Biology, Physiological Botany, Linnean Centre for
Plant Biology, Uppsala University, Uppsala, Sweden, 5 Department of Plant Biotechnology and
Bioinformatics, Ghent University, Ghent, Belgium, 6 Center for Plant Systems Biology, Ghent, Belgium,
7 Department of Medical Cell Biology, Uppsala University, Uppsala, Sweden, 8 Institute of Molecular
Physiology and Biotechnology of Plants, University of Bonn, Bonn, Germany, 9 Laboratoire Reproduction et
Développement des Plantes, ENS de Lyon, Université Claude Bernard Lyon 1, CNRS, INRAE, Lyon, France,
10 Department of Plant Nutrition, Institute of Crop Science and Resource Conservation, University of Bonn,
Bonn, Germany

* Panagiotis.moschou@uoc.gr

California San Diego, UNITED STATES

**Data Availability Statement:** All relevant data are
within the paper and its Supporting Information
files.

## Abstract

Protein function can be modulated by phase transitions in their material properties, which
can range from liquid- to solid-like; yet, the mechanisms that drive these transitions and
whether they are important for physiology are still unknown. In the model plant Arabidopsis,
we show that developmental robustness is reinforced by phase transitions of the plasma
membrane-bound lipid-binding protein SEC14-like. Using imaging, genetics, and in vitro
reconstitution experiments, we show that SEC14-like undergoes liquid-like phase separa-
tion in the root stem cells. Outside the stem cell niche, SEC14-like associates with the cas-
pase-like protease separase and conserved microtubule motors at unique polar plasma
membrane interfaces. In these interfaces, SEC14-like undergoes processing by separase,
which promotes its liquid-to-solid transition. This transition is important for root develop-
ment, as lines expressing an uncleavable SEC14-like variant or mutants of separase and associ-
ated microtubule motors show similar developmental phenotypes. Furthermore, the pro-
cessed and solidified but not the liquid form of SEC14-like interacts with and regulates the
polarity of the auxin efflux carrier PINFORMED2. This work demonstrates that robust devel-
opment can involve liquid-to-solid transitions mediated by proteolysis at unique plasma
membrane interfaces.

**Funding:** Funding for this work was through the Vetenskapsrådet (VR) (298264-2015 to PNM), Svenska Forskningsrådet Formas (MOP-86675 to PNM), Hellenic Foundation for Research & Innovation (HFRI)-Always Strive for Excellence-Theodore Papazoglou (1624 to PNM), Hellenic Foundation of Research and Innovation (HFRI) (06526 to AM), National Secretariat of research and innovation (GR) (T2Δ-00597 to PNM), H2020 Marie Skłodowska-Curie Actions (RISE 872969 PANTHEON to PNM), Foundation for Research and Technology (FORTH-IMBB) Start-Up Funding (to PNM), and by the Deutsche Forschungsgemeinschaft (SCHA 1274/5-1, 841 Germany's Excellence Strategy EXC-2070-390732324 PhenoRob to GS). The funders had no role in study design, data collection and analysis, decision to publish, or preparation of the manuscript.

**Competing interests:** The authors have declared that no competing interests exist.

**Abbreviations:** APM, amiprophos-methyl; BFA, brefeldin A; CIDER, classification of intrinsically disordered ensemble regions; DRP, Dynamin-related protein; ESP, EXTRA SPINDLE POLES; FRAP, fluorescence recovery after photobleaching; FRET, Förster resonance energy transfer; GST, glutathione S-transferase; IDR, intrinsically disordered region; KIN7, Kinesins 7-clade; KISC, kinesin-separase complex; LLPS, liquid–liquid phase separation; LUV, large unilamellar vesicle; MS, mass spectrometry; MT, microtubule; NCPR, net charge per residue; PCC, Pearson correlation coefficient; PE, phosphatidylethanolamine; PI, phosphatidylinositol; PIN, PINFORMED; PLA, proximity ligation assay; PLAAC, prion-like amino acid composition; PLD, prion-like domain; PM, plasma membrane; QCM-D, quartz crystal microbalance with dissipation; rBiFC, ratiometric bimolecular fluorescence complementation; *rsw4*, *radially swollen 4*; RT, room temperature; SFH8, SEC FOURTEEN-HOMOLOG8; TGN, *trans*-Golgi network; TIRFM, total internal reflection fluorescence; Y2H, yeast two-hybrid.

## Introduction

Under certain conditions, biomolecules can separate from their bulk phase through liquid–liquid phase separation (LLPS), thereby attaining liquid-like properties, such as surface tension, which leads to highly circular condensates akin to droplets [1]. LLPS determines the formation of many evolutionary conserved condensates, such as nucleoli, stress granules, and processing bodies. Starting as liquids, some condensates undergo transitions in their material properties that affect their viscosity, surface tension, and degree of penetrance by other molecules. For example, in *Drosophila melanogaster*, *oskar* ribonucleoprotein (RNP) condensates undergo a liquid-to-solid transition, which is important for the polar distribution of some RNAs in the cell [2]. Whereas *oskar* RNP liquidity allows RNA sequestration, its solid phase precludes the incorporation of RNA while still allowing protein sequestration. Furthermore, although they are not delimited by membranes, condensates can interface with them or even engulf small vesicles [3].

The past few years have experienced tremendous progress in the evolution of a molecular grammar that underpins LLPS. Molecules such as proteins and RNAs are polymers with attractive groups known as "stickers" that form noncovalent and mainly weak interactions. At certain concentrations, which are determined by various factors (e.g., temperature, redox state, pH), interactions are enabled among intra- or intermolecular stickers. When reaching a system-specific threshold concentration, the whole system containing various proteins and/or RNAs undergoes LLPS. The stickers promote the attraction between charged residues, dipoles, or aromatic groups that are usually provided by the so-called "intrinsically disordered regions" (IDRs) [4]. Stickers are connected by "spacers" that regulate the density transitions (i.e., liquid-to-solid transitions) by orienting stickers. The IDRs lack a defined structure and thus can easily expose their stickers. Furthermore, IDRs can increase the apparent size known as hydrodynamic radius adopted by the solvated, tumbling protein molecule [5].

In the model plant, Arabidopsis (*Arabidopsis thaliana*) LLPS condensates are involved in, for example, the internal chloroplast cargo sorting, transcriptional circuits modulating defence, RNA processing, and temperature sensing [6–10]. Furthermore, plants form conserved condensates like stress granules and processing bodies [11–13]. Recent evidence suggests that like their animal counterparts, plant condensates can interface with membranes. For example, condensates of the TPLATE, a plant-specific complex modulating endocytosis, can likely form on the plasma membrane [14]. We have also shown that condensates of processing bodies form on membranes in Arabidopsis and can attain polarity (i.e., localizing asymmetrically at the plasma membrane) [13]. However, the functional significance of condensates at the plasma membrane is unclear.

In plants, the few known polar plasma membrane proteins provide crucial information for robust development [15–17]. We have previously discovered a link between development and a complex comprising the Arabidopsis caspase-like protease separase (also named EXTRA SPINDLE POLES [ESP]) and 3 Arabidopsis microtubule (MT)-based centromeric protein-E-like Kinesins 7 (KIN7), which belong to the so-called KIN7.3-clade (KIN7.1, KIN7.3, and KIN7.5). This complex (the kinesin-separase complex [KISC]) is recruited to MTs; the most abundant and important kinesin from the KISC is KIN7.3 [18]. ESP is an evolutionarily conserved protein responsible for sister chromatid separation and membrane fusion in both plants and animals [19,20]. ESP binds to the KIN7.3-clade C termini (the so-called "tails"), inducing conformational changes that expose the MT-avid N-terminal motor domain of KIN7s, thereby increasing KISC binding on MTs. The KISC can also modulate polar domains of the plasma membrane (PM), as the temperature-sensitive *radially swollen 4* (*rsw4*) mutant harbouring a temperature-sensitive *ESP* variant or KIN7.3-clade mutants display reduced delivery of polar

auxin efflux carriers PINFORMED (PINs) at the PM [18]. Yet, how the KISC acts upon PM polar domains to regulate development remains elusive.

Whether condensates interfacing with membranes can undergo liquid-to-solid transitions like cytoplasmic ones and if these transitions would have any significance is unclear. Here, we discovered that a previously uncharacterized SEC14-like lipid transfer protein that we named SEC FOURTEEN-HOMOLOG8 (SFH8) recruits KISC to the PM. The ESP part of KISC trimmed SFH8 protein removing an IDR, leading to the conversion of SFH8 from a liquid to a more solid filamentous phase that remains attached to the PM, an event that we could also reconstitute in vitro. This liquid-to-solid transition was associated with SFH8 polarization, interaction with PIN2, and robust root development. Remarkably, we showed how spatiotemporally confined proteolysis can yield changes in the material properties of proteins and how these underlie robust development.

## Results

### The KISC associates with the lipid-transfer protein SFH8 at polar PM domains

As the KISC regulates processes that are relevant to the PM (e.g., PIN delivery), we aimed to survey an underlying molecular mechanism. We observed that in the distal meristem of the root (as defined below), ESP and KIN7.3, detected by native antibodies, decorated the PM at apical domains in the outermost layer, the epidermis and basal domains in the adjacent layer, the cortex (**S1A and S1B Fig**). We obtained similar results with ESP and KIN7.3 fluorescent fusions, under an estradiol-inducible module driven by the KIN7.3 promoter (for ESP) or the meristem-specific promoter RPS5a (for KIN7.3) (**S1C–S1E Fig**). As KISC proteins lack lipid-binding motifs, we postulated that the KISC associates with the PM via a protein tether, which we sought to identify by screening a yeast two-hybrid (Y2H) library using KIN7.3 as bait (**S2A Fig**). Among the 5 clones identified, we focused on AT2G21520, as its encoded protein showed localization reminiscent of the PM when expressed transiently in *Nicotiana benthamiana* leaves (**S2B Fig**; **S1 Text**). The protein encoded by AT2G21520 is a SEC14-like protein (BLAST-P: $p = 1 \times 10^{-49}$) that was ascribed the symbol SFH8, bearing a C terminal "nodulin"-like motif (aa 479–637) punctuated with positively charged lysine (K) residues (**S2C Fig and S1 File**) [21]. This nodulin-like motif promoted the interaction with KIN7.3 tail (**S2A Fig**). SFH8 is a genuine SEC14-like protein, as it could rescue the *Saccharomyces cerevisiae sec14-1* temperature-sensitive mutant (**S2D Fig**; 34.5˚C) [22]. We confirmed that GFP-KIN7.3 likely interacts with FLAG-tagged SFH8 in stable Arabidopsis lines, as shown by co-immunoprecipitation (**Fig 1A**). Furthermore, SFH8 associates with KIN7.3 likely at the PM, as evidenced by a transient ratiometric bimolecular fluorescence complementation (rBiFC) assay in Arabidopsis root protoplasts (**Fig 1B**). Unlike conventional BiFC, which lacks an internal reference marker, rBiFC can distinguish weak interactions from background fluorescence levels [23]. In rBiFC, we used as a positive control the KIN7.3 interaction with the N terminus of ESP (aa 1–791; "DomA"), while as negative the KIN7.3 interaction with the C terminus of ESP (aa 1,622–2,178; "DomC") [18].

The localization and functions of SFH proteins in Arabidopsis are unknown; genetic evidence suggests that SFH1 is essential for root hair development [21]. We expressed *SFH8-mNeon* under the *SFH8* promoter (*SFH8pro*) to explore its localization; the SFH8-mNeon signal was exclusively observed in the root meristem. To expedite our localization analyses, we defined 4 developmental root regions along the proximodistal axis: core meristem (1; stem cell niche); meristematic zone (2; proximal meristem); meristematic/transition zone (3); and late transition zone (distal meristem; root regions described in **Fig 1C**) (4). We

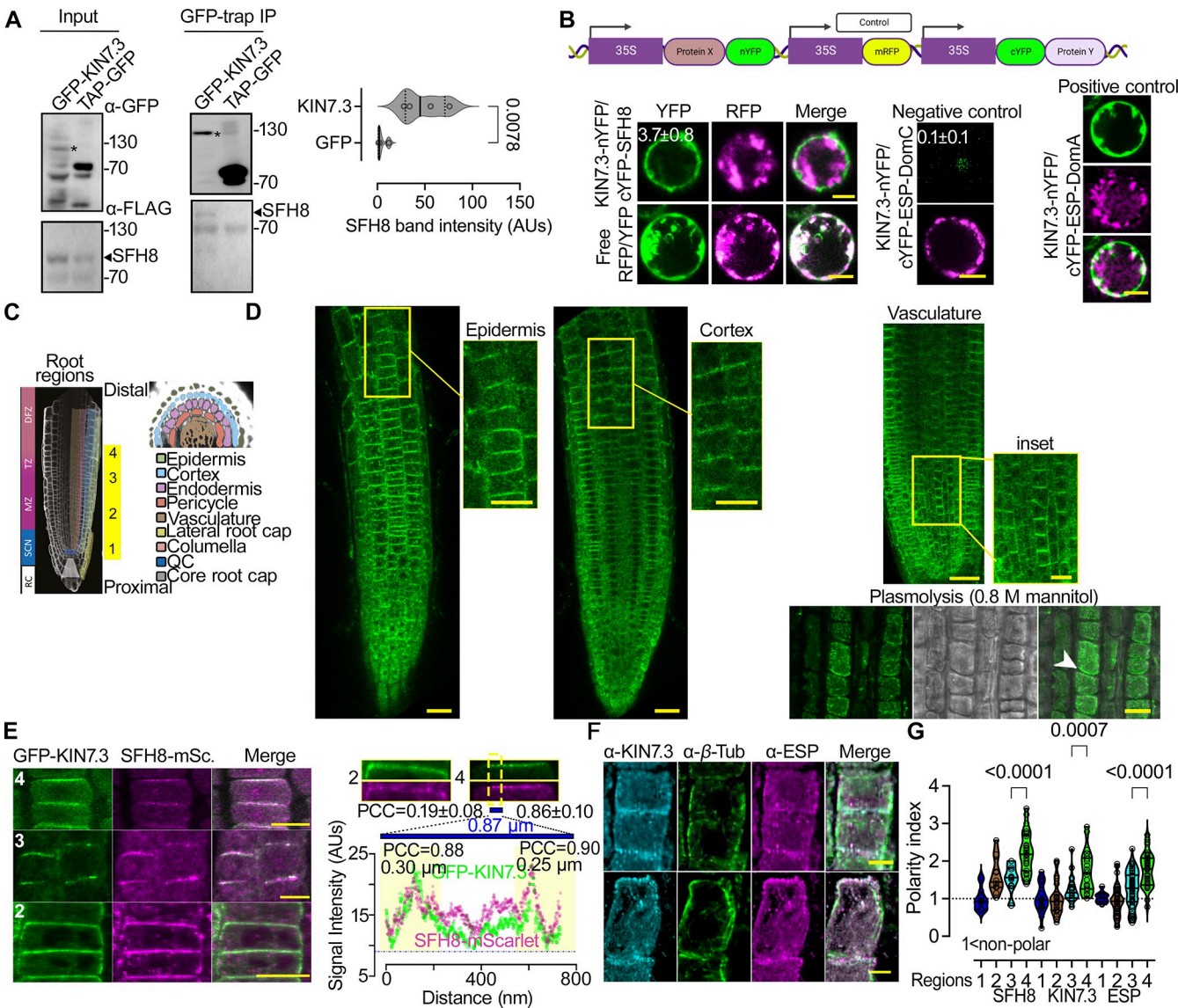

**Fig 1. KISC associates with the lipid-transfer protein SFH8 at the PM.** (**A**) Co-immunoprecipitation and immunoblots from Arabidopsis seedlings coexpressing *KIN7.3pro:KIN7.3-GFP* with either *RPS5apro:3xFLAG-SFH8* or *2x35Spro:TAP-GFP* (5 DAG). Right: quantification of the interaction (SFH8 signal intensity detected by α-FLAG that was pulled down by GFP or GFP-KIN7.3; N = 4 pooled experiments, n = 1 assay; p-value was calculated by a 2-tailed t test). Asterisks in the immunoblots denote the full-length GFP-KIN7.3, which is sensitive to proteolytic degradation in the input sample. (**B**) Ratiometric transient BiFC assays in root protoplasts (the cartoon on the top shows the construct used). Controls: KIN7.3-nYFP coexpressed with ESP truncations known as DomA (1–791; positive control) or DomC (1,622–2,178; negative control) as defined previously [18]. The mean YFP/RFP signal ratios ± SD are indicated on images (N = 2 pooled experiments, n = 15 cells). Scale bars, 6 μm. (**C**) Root model showing the "4 root regions" examined herein: SCN (1); MZ (2); TZ (3); DFZ (4). (**D**) Tissue-specific expression and subcellular localization of SFH8-mNeon in roots (*SFH8pro:SFH8-mNeon* expressing lines; 5 DAG, at the indicated tissues). The plasmolysis experiment confirms SFH8 signal exclusion from the cell wall (note the white arrowhead; region 2). The experiment was replicated 5 times. Scale bars, 20 μm (5 μm in the insets or plasmolysis experiment). (**E**) Colocalization of GFP-KIN7.3 (KIN7.3pro) and SFH8-mScarlet (SFH8pro; epidermis regions 1–4). Scale bars, 10 μm. Right top: high-resolution signal of KIN7.3/SFH8 at the PM (epidermis regions 2, 3, and 4). The overall PCC values for regions 2 and 4 are shown (ROIs: whole image). For region 4, a plot profile of signal intensity across a straight line of 0.87 μm and 2 peak PCC regions (0.30 and 0.25 μm) are shown. (Data are means ± SD, N = 3 pooled experiments, n = 3 adjacent cells per experiment.) (**F**) Example of α-ESP/α-KIN7.3 colocalization and polarization (counterstained with α-β-Tubulin; epidermis, region 3). Scale bars, 5 μm. (**G**) SFH8, KIN7.3, and ESP polarity index in regions 1–4 (values >1 denote polarization; polarity index calculation is described in **S1A Fig**; data are means ± SD, N = 5 pooled experiments, n ≥ 12 cells per experiment; p-values were calculated by 1-sided Dunnett). Raw data can be found in the Supporting information section (S1 Data and S1 Raw Images). AUs, arbitrary units; BiFC, bimolecular fluorescence complementation; Co., cortex; DAG, day after germination; DFZ, differentiation zone; Ep., epidermis; ESP, EXTRA SPINDLE POLES; KISC, kinesin-separase complex; mSc., mScarlet; MZ, meristematic zone; PCC, Pearson correlation coefficient; PM, plasma membrane; QC, quiescent center; RC, root columella; ROI, region of interest; SCN, stem cell niche; SFH8, SEC FOURTEEN-HOMOLOG8; TAP, tandem affinity purification tag; Tub., tubulin; TZ, transition zone.

detected SFH8-mNeon signals in all meristematic root cells, at apical PM domains in the epidermis and basal domains in the cortex/vasculature in distal meristem cells (**Fig 1D**), like the KISC proteins (**S1B–S1E Fig**). Accordingly, SFH8 colocalized at the PM with KIN7.3, mainly in regions 3 and 4, as revealed by analysis of signal collinearity in super-resolution micrographs (120 nm) using the Pearson correlation coefficient (PCC) to quantify colocalization (**Fig 1E**, right chart and below). Furthermore, SFH8 and KISC proteins attained significant and similar polarity in regions 3 and 4, localizing to basal (in the cortex) or apical domains (epidermis) (**Fig 1E–1G** for SFH8). Later, we discuss this polarization in more detail, but altogether, these results suggest that KISC proteins associate with SFH8 at polar domains of the PM in root cells.

## SFH8 clusters recruit the KISC where ESP cleaves SFH8 creating filaments

The interactions between KISC-SFH8 prompted us to examine whether SFH8 is tethering KISC at the PM. To address this question, we identified 2 T-DNA insertion mutants in *SFH8*, designated *sfh8-1* and *sfh8*-2 (**Fig 2A**). We continued further analyses with the *sfh8-1* mutant (hereafter "*sfh8*") because as explained later, *sfh8-1* phenotype is similar to *sfh8*-2. In *sfh8*, GFP-KIN7.3 displayed both a reduced PM localization and polarity compared to that in the wild type (**Fig 2B**). SFH8-mNeon tethering at the PM did not appear to depend on KISC, as SFH8 could still be tethered at the PM in the 2 partial loss-of-function KISC mutants, the KIN7.3-clade mutant *kin7.1 kin7.3 kin7.5* (*k135*; [18]) and *rsw4* backgrounds (**Fig 2C**). Interestingly, in all cell types of *k135* or *rsw4* examined, SFH8-mNeon was apolar (**Fig 2C** and approximately 2-fold difference in polarity index as defined in **S1A Fig**). To further validate this result, we used an inducible system that leads to the overaccumulation of the KIN7.3 C terminal tail with the ability to deactivate KISC (*XVEpro>KIN7.3pro:HA-KIN7.3tail*), as it titrates ESP out of the active KISC [18]. Thus, this transient depletion led to a loss of SFH8 polarity and perturbed gravitropism within 2 days, as expected given the link of KISC to PINs that regulate auxin and gravitropism (**Fig 2D–2F** and [18]). Hence, KISC and SFH8 synergistically define their localization: SFH8 tethers KISC at the PM, and, in turn, KISC promotes SFH8 polarization.

Interestingly, in follow-up experiments aiming at studying in detail the localization of SFH8 in *k135* and *rsw4* backgrounds, we observed that the full-length SFH8 levels increased in these 2 mutants (**Fig 3A**, "FL" arrowhead). In particular, immunoblot analysis of lines expressing a construct encoding SFH8 with a hexahistidine-triple-flag (referred to as HF)-mScarlet tag at the C or N terminus (approximately 106 kDa) under the control of the *RPS5apro* showed increased full-length SFH8 abundance in *k135* and *rsw4* backgrounds (**Fig 3A**). We used the *RPS5apro* here as the *SFH8pro* could not lead to a detectable signal in immunoblots. Given the increased abundance in *k135* and *rsw4* backgrounds of SFH8 and considering that ESP is a protease, we decided to examine the possibility that KISC regulates SFH8 levels. The reduced abundance of SFH8 levels in the wild type compared to that in *k135* and *rsw4* associated with a presumptive approximately 40-kDa (or approximately 10-kDa excluding mScarlet) N-terminal cleavage product (**Fig 3A**, arrowhead with asterisk). When SFH8 was tagged C-terminally with HF, it produced a double band, consistent with the cleavage of SFH8 at the N terminus (**Fig 3A**, right blot). We, thus, speculated that in the presence of KISC, SFH8 is cleaved from ESP close to its N terminus, producing a 10-kDa product (hereafter, identified as "cleavage product"). We followed up the putative SFH8 cleavage in vivo using lines with *SFH8* tagged C- or N-terminally with the mNeon fluorescent protein (**Fig 3B**). We observed fluorescent cytoplasmic puncta that accumulated gradually from regions 1 to 4 in mNeon-SFH8 expressing lines in regions 3 and mainly in 4 that, as shown above, KISC shows strong colocalization with

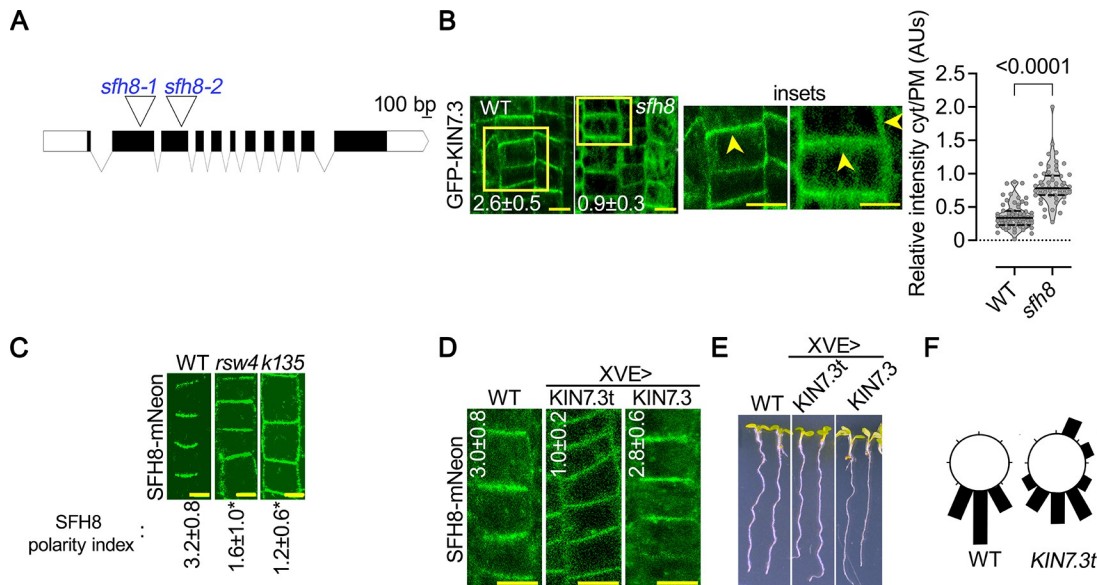

**Fig 2. SFH8 recruits KISC at the PM and KISC regulates SFH8 polarity.** (**A**) T-DNA insertion sites for *sfh8–1* and *sft8–2* (second and third exons, respectively). (**B**) GFP-KIN7.3 PM localization in WT or *sfh8* (5 DAG, epidermis region 3). The polarity index of KIN7.3 is also shown on the images (data are means ± SD, $N$ = 3 pooled experiments from region 3, $n$ = 6–8 cells per experiment; differences were significant at $p < 0.0001$ and calculated by a 1-sided Dunnett). Arrowheads in the insets show apical or lateral localization of KIN7.3. Scale bars, 5 μm. Right: quantification of cytoplasmic to PM signal (data are means ± SD, $N$ = 3 pooled experiments, $n$ = 18–24 cells per experiment; "*": $p < 0.0001$ to WT, calculated by a 2-tailed $t$ test). (**C**) SFH8-mNeon (SFH8pro) localization in WT, *rsw4*, and *k135* (5 DAG, region 3). Images are representative of an experiment replicated >10 times for polarity. Numbers in micrographs are the polarity indexes of SFH8 (data are means ± SD, $N$ = 3 pooled experiments from region 3, $n$ = 9–17 cells per experiment; "*": $p < 0.0001$ to WT, calculated by a 1-sided Dunnett). Scale bars, 5 μm. (**D**) SFH8-mNeon (SFH8pro) polarity loss in lines overexpressing transiently KIN7.3 full length or KIN7.3 tail ("t"; *KIN7.3pro>XVEpro* module induced for 24–36 h with 2 μM estradiol; epidermis region 3). Numbers in micrographs are the polarity indexes of SFH8 (Data are means ± SD, $N$ = 3 pooled experiments from region 3; $n$ = 410; "*": $p < 0.0001$ to WT, calculated by a 1-sided Dunnett). Scale bars, 10 μm. (**E**) Perturbed gravitropism and growth of lines overexpressing transiently full-length or tail KIN7.3 ("t"; *KIN7.3pro>XVEpro* module induced for 24–36 h with 2 μM estradiol). Scale bars, 8 μm. (**F**) Circular plots showing the quantification of perturbed gravitropism in *KIN7.3pro>XVEpro>KIN7.3t* expressing roots (data are means ± SD, $N$ = 3 pooled experiments, $n$ = 8–10 roots per experiment). Raw data can be found in the Supporting information section (S1 Data). DAG, day after germination; KISC, kinesin-separase complex; PM, plasma membrane; *rsw4*, *radially swollen 4*; SFH8, SEC FOURTEEN-HOMOLOG8; WT, wild type.

SFH8; these foci were absent from the C-terminally tagged mNeon lines (SFH8-mNeon), where the SFH8 signal was mostly on the PM (**Fig 3B**), suggesting that likely only the N-terminus of SFH8 is cleaved and released in the cytoplasm. These results altogether suggest that SFH8 is progressively cleaved during development by ESP at the N terminus part, creating a cleavage product in the form of cytoplasmic puncta.

To further dynamically follow SFH8 cleavage in vivo, we established a double-labelled N-terminal/C-terminal tagged fluorescent SFH8 (hereafter "cleavage biosensor"). We speculated that mScarlet-SFH8-mNeon cleavage would disrupt the colocalization of mNeon and mScarlet signals. Indeed, we observed a lack of mNeon/mScarlet colocalization in region 3 and mainly in 4 (**Fig 3C**). High-resolution imaging at the PM defined a more clustered form of SFH8 in region 1 (where both signals colocalize, indicative of an intact cleavage biosensor) and a more filamentous form in regions 3 and 4 of the remaining C terminal part of the SFH8 (where no colocalization between mNeon/mScarlet is observed) (**Fig 3C,** detail and graph). We thus showed that SFH8 transitions from a cluster (full-length protein) to a filament (containing only the C terminal part) upon its cleavage from ESP (**Fig 3C**, compare "filaments" versus "clusters").

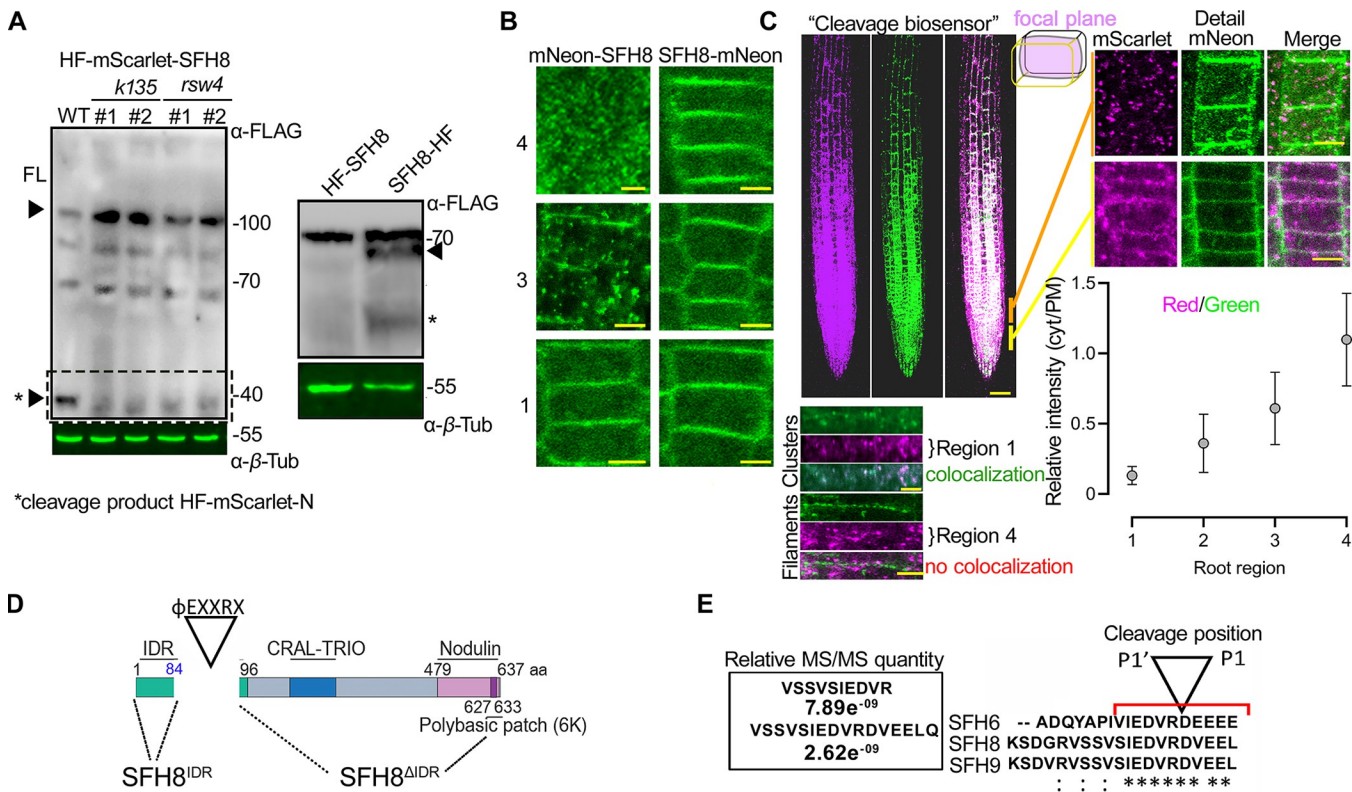

**Fig 3. KISC trimming of SFH8 promotes the cluster-to-filamentous transition.** (**A**) Detection of SFH8 N terminal fragment from lines expressing *HF-mScarlet-SFH8* (RPS5apro) (black arrowhead with an asterisk at approximately 40 kDa) in WT, *k135*, or *rsw4* backgrounds (24 h at the restrictive temperature 28°C to induce *rsw4* mutation [28]; 2 lines were used). Right: immunoblot showing the remaining C terminal SFH8 fragment (black arrowhead; asterisk shows an additional truncated product of low abundance) in WT lines expressing HF-SFH8 (RPS5apro). The experiment was replicated 4 times. (**B**) Localization of *SFH8* N- or C-terminally tagged with mNeon (SFH8pro; 7 DAG, epidermis of regions 1, 3, and 4). Note the formation of puncta in the cytoplasm of lines expressing *mNeon-SFH8* at region 3 onwards and the reduction of the corresponding PM signal for mNeon-SFH8. The experiment was replicated 5 times. Scale bars, 3 μm. Images are representative of an experiment replicated >10 times. (**C**) Localization of an SFH8 cleavage biosensor (*RPS5apro:HF-mScarlet-SFH8-mNeon*). Scale bar, 50 μm. Upper right: details of regions 1 and 3 (mid-plane epidermis, scale bars, 4 μm), and relative signal intensity of cytoplasmic versus PM signal (chart). Data are means ± SD (*N* = 10 pooled experiments, *n* = 4 cells per experiment). Lower panel (left): super-resolution imaging of cluster-to-filament conversion (epidermis regions 2 and 4). Note the absence of mScarlet signal from filaments (denoted as "no colocalization"). The experiment was replicated 5 times. Scale bars, 0.8 μm. (**D**) SFH8 protein architecture (IDR corresponding to aa 1–96; CRAL-TRIO: active site for SEC14 proteins). The φEXXR cleavage motif for ESP is also shown. (**E**) SFH8 IDR peptides identified in *pRPS5a:SFH8-mScarlet-HF* pull-down experiments coupled with LC–MS/MS. Right: the cleavage motif of ESP on SFH proteins, φEXXR is conserved (presented here for 3 SFH protein paralogs, SFH6/8/9). P1'-P1 correspond to residues R and D, respectively. Raw data can be found in the Supporting information section (S1 Data and S1 Raw Images). DAG, day after germination; FL, full-length; IDR, intrinsically disordered region; KISC, kinesin-separase complex; PM, plasma membrane; *rsw4*, *radially swollen 4*; SFH8, SEC FOURTEEN-HOMOLOG8; WT, wild type.

We also aimed at defining the exact cleavage site within SFH8. Accordingly, we immunoprecipitated SFH8-mScarlet-HF using α-FLAG and quantified the abundance of SFH8 peptides via mass spectrometry (MS), resulting in the identification of a potential cleavage site right after the residue R84 (**Fig 3D and 3E**). The size of the predicted cleavage fragment was in good agreement with the immunoblots shown in **Fig 3A** (approximately 10 kDa). The $I^{80}EDVR^{84}D$ sequence corresponded to the reported non-plant ESP cleavage consensus motif φEXXR [24,25], also found in other SFH8-like proteins (**S1 File**). By establishing an in vitro ESP cleavage assay, we confirmed that immunopurified ESP, mitotically activated through coexpression with *Cyclin D* [26], can cleave recombinant glutathione *S*-transferase (GST)-SFH8 at R84; we validated our assay by showing the cleavage of a cohesin (SYN4), the well-

known target of ESP (**S3A–S3D Fig**) [27]. Hence, the filamentous conversion of SFH8 to filaments likely depends on the cleavage at R84.

## SFH8 forms transient clusters with liquid-like properties that exclude KISC

We further aimed to follow the colocalization of KISC with SFH8 and the cleavage of SFH8 in more detail. As the KISC binds MTs [18], and since SFH8 showed a filamentous localization in regions 3 and 4, we postulated that SFH8 and the KISC might copartition in MT filaments in proximity to the PM. Contrary to our expectations, the marker MAP4$^{MBD}$ (MT-binding domain of MICROTUBULE-ASSOCIATED PROTEIN4) or $\beta$-tubulin showed only partial colocalization with KIN7.3 at the PM at less than 10% of the filaments (**S4A Fig**). Furthermore, amiprophos-methyl (APM) that disassembles MTs (10 nM; [18]) did not significantly alter KIN7.3 localization at the PM, although a small part of KIN7.3 filaments and, in particular, their edges remained attached in some cases in bundled MTs (**S4B Fig**; approximately 10%). In Arabidopsis roots, SFH8 filaments were short (<0.5 μm) and insensitive to APM treatment; ESP decorated similar filaments as shown in root cells expressing GFP-tagged ESP under an estradiol-inducible promoter driving expression at KIN7.3 domains (*KIN7.3-pro>XVEpro>GFP-ESP/RPS5apro:SFH8-mScarlet*; **S4B and S4C Fig**). Actin depletion through latrunculin B also did not alter SFH8 localization or clustering at the PM in lines coexpressing LifeAct-mCherry with mNeon-SFH8 (**S4D Fig**). Furthermore, SFH8 did not colocalize with actin filaments at the PM (**S4D Fig**, right). These results suggest that SFH8 and KISC do not remain attached to MTs or actin at the PM.

As the previous results suggested that the KISC and SFH8 coassemble in cytoskeleton-independent filaments, we aimed at deciphering KISC and SFH8 localization in detail at the PM. We thus examined the localization of KISC components and SFH8 in Arabidopsis roots by total internal reflection fluorescence microscopy (TIRFM), which is suitable for analyzing the PM due to the shallow illumination penetration. By focusing on lateral cell junction domains (**Fig 4A**; regions 3 and 4; 3 to 5 days after germination), we determined that SFH8-mNeon segregates into at least 2 major populations: (i) immobile filaments that colocalize with KIN7.3-tagRFP and (ii) mobile or immobile KIN7.3-tagRFP-independent cluster-like structures (**Fig 4B–4E**). These results are consistent with the above observation of SFH8 and KISC localization in clusters and filaments (**Fig 3C**). The mobile SFH8 clusters showed little diffusion at the PM, circularity, and occasionally fused (or underwent fission), properties that are reminiscent of cellular condensates that sometimes form through LLPS (**Fig 4C and 4D** and **S1–S4 Movies**). We also observed some small nondiffusing clusters with reduced circularity (**S3 and S4 Movies**) that may show intermediate phases between the cluster state (droplet-like) and the filamentous state. As a cautionary note here, we did not examine other parameters used for cytoplasmic condensates, such as dripping or saturation concentrations as membrane-bound condensates, deform through the physical interfacing with the underlying lipids (the process known as wetting; [29–32]).

KIN7.3 and SFH8 colocalized in short filaments but not in SFH8-decorated clusters; these clusters showed variable residence times at the PM, unlike filaments that were permanently assembled at the PM (**Fig 4F and 4G**). We wished to determine why the KISC was excluded from the SFH8 clusters; we hypothesized that converting the polybasic charge of the SFH8 nodulin patch to a hydrophobic region would promote the clustered (condensed) state of SFH8. This hypothesis is based on the counterion negative charge at the PM that could attenuate repulsion of positively charged residues at the nodulin part of SFH8 (suggested previously for membrane-associated peptides [32]). Indeed, replacing 6 pertinent lysines (K) with alanines (A; SFH8$^{6KtoA}$) in the nodulin patch artificially increased SFH8 clustering in *N*.

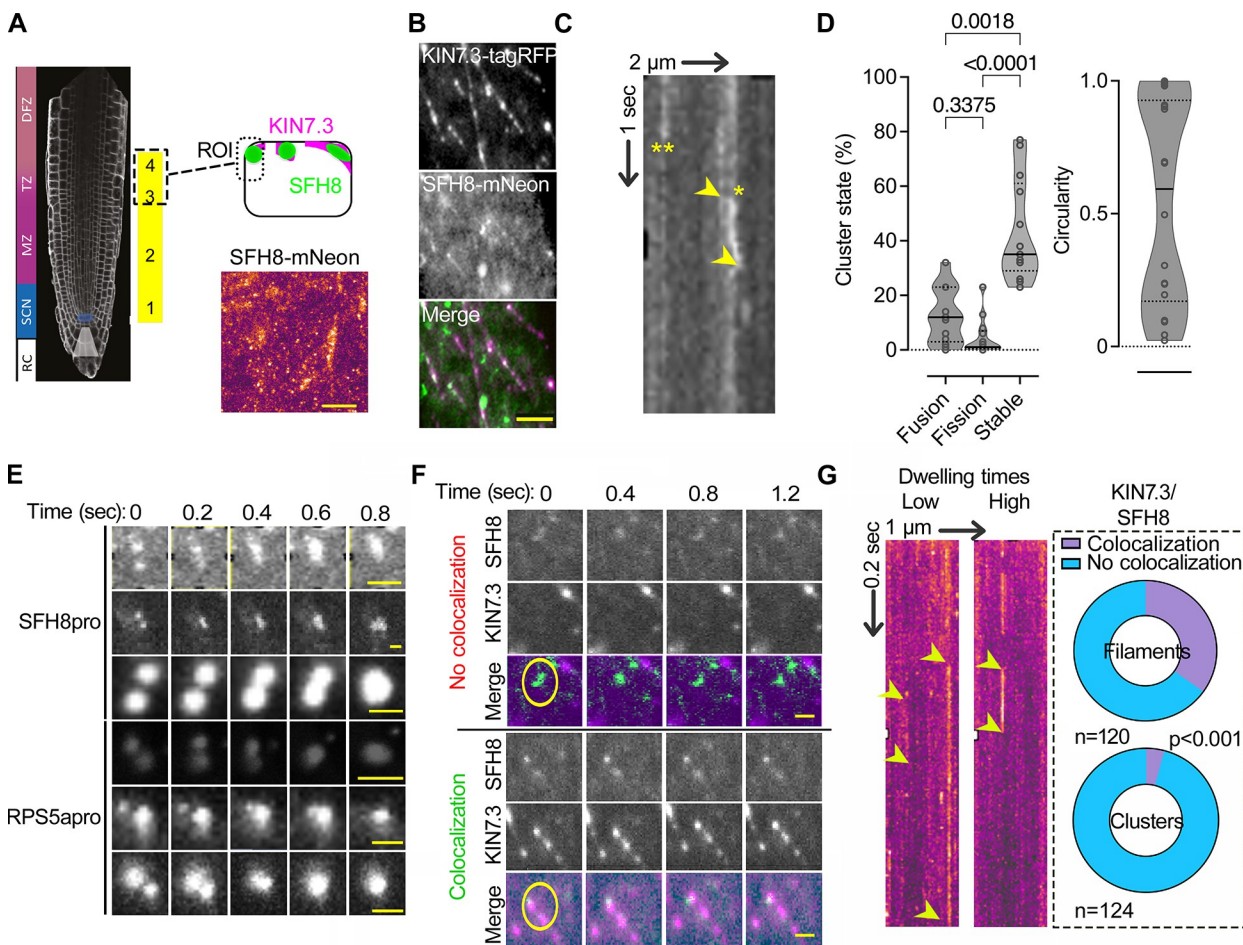

**Fig 4. SFH8 forms PM liquid-like clusters that lack association with KISC.** (A) TIRFM setting for visualization of SFH8 at the PM. The model is showing the region used for imaging, and an example TIRFM micrograph of SFH8-mNeon (lower right; SFH8pro). Scale bar, 2 μm. In TIRFM imaging, the focal plane is restricted to the outermost tissues, and, therefore, epidermis of region 1 or 2 is inaccessible (see **Fig 1C** for a root model showing that the epidermis in this region is encapsulated by the root cap). (**B**) Example of a dual-channel TIRFM of lines expressing SFH8-mNeon (SFH8pro) and KIN7.3-RFP (RPS5apro). The experiment was replicated 5 times. Scale bar, 0.3 μm. (**C**) Kymograph showing laterally diffusing (*) and nondiffusing (**) clusters of SFH8. Arrowheads indicate the spatial offset of the diffusing cluster (lateral displacement on the PM plane is around approximately 200 nm). The arrows (2 μm and 1 s) show the spatiotemporal resolution. (**D**) Quantification of SFH8 clusters (*SFH8pro*) in 3 different stages, fusion, fission, and "stable" (i.e., not undergoing fission or fusion). The circularity of clusters is also shown (right). Data are means ± SD (N = 3 pooled experiments, n = 4–6 fields with percentages per experiment; the *p*-values were calculated by 1-way ANOVA). (**E**) Examples of SFH8-mNeon clusters fusing on the PM. Note that similar sizes and fusion dynamics of clusters were observed with 2 promoters (SFH8pro and RPS5apro), suggesting independence of these parameters from expression levels (higher for RPS5apro). Scale bars, 0.3 μm. (**F**) Dual-channel TIRFM of SFH8-mNeon/KIN7.3-RFP coexpressing line showing SFH8 clusters and the formation of filaments that do not diffuse. Note the lateral diffusion of SFH8 clusters and the lack of filaments motility (circles). The experiment was replicated 3 times. Scale bars, 0.3 μm. (**G**) Kymographs show clusters with low (left) and high (right) dwelling times at the PM. Right: pie graphs showing quantifications of KIN7.3 and SFH8 colocalization percentages in clusters or filaments (N = 3 pooled experiments, n = as indicated; *p*-values were calculated by Wilcoxon). The arrows (1 μm and 0.2 s) show the spatiotemporal resolution. Raw data can be found in the Supporting information section (S1 Data). KISC, kinesin-separase complex; PM, plasma membrane; SFH8, SEC FOURTEEN-HOMOLOG8; TIRFM, total internal reflection fluorescence microscopy.

*benthamiana* leaves (although also exhibiting a slightly reduced localization at the PM) and decreased its association with KIN7.3 in Y2H (**S4E and S4F Fig**). In Arabidopsis, mNeon-SFH8$^{6KtoA}$ showed reduced localization at the PM, reduced filaments, and lacked polarity (**S4G Fig**). This result suggested that hindering the interaction between SFH8 and the KISC by reducing the accessibility to the SFH8 blocked SFH8 filamentous transition. Hence, filaments

are produced through KISC where KISC-SFH8 remain associated. On the other hand, the liquid-like SFH8 clusters are not accessible by KISC.

## The N-terminus of SFH8 defines its liquid-like properties

To address the link between the removal of the SFH8 N-terminus by KISC and changes in SFH8 structure at the PM (i.e., the filamentous transition), we first aimed at linking SFH8 lack of diffusion with SFH8 filaments (considering their likely permanent residence at the PM). We thus used fluorescence recovery after photobleaching (FRAP) to follow SFH8 diffusion. Owning to their transient association of SFH8 clusters with the PM as shown by TIRFM, we anticipated that PMs with liquid-like SFH8 clusters would show increased FRAP rates. Indeed, SFH8-mNeon showed recovery at the PM close to the meristem (regions 1 and 2), unlike the distal meristem (regions 3 and 4) in which SFH8 lacked recovery (**Fig 5A**). As this filamentous transition of SFH8-mNeon was also reduced in the KISC mutants (**Fig 5B**; >3-fold), these findings further genetically confirm that KISC mediated the conversion of SFH8 clusters to solid-like filaments. These filaments are more stably attached to the PM as they do not show recovery in FRAP (**Fig 5A**), which is consistent with the lack of mobility in TIRFM (see above, **Fig 4**), and can retain an association with the KISC (**Figs 4 and 5A**, model). Hence, FRAP confirmed that SFH8 filaments are stably attached to the PM and that cleavage by KISC could be somehow involved in this cluster-to-filament transition.

Next, we asked whether SFH8 cleavage fragment removal associates with the cluster-to-filament transition. Through in silico predictions, we determined that the cleavage fragment is an IDR (hereafter SFH8$^{IDR}$; **Fig 5C**). We established that this protein architecture is conserved throughout the evolution of SFH proteins, which implies the functional importance of this IDR (**S2 File**). As IDRs are usually enriched in proteins undergoing LLPS [33] and considering the liquid-like behaviour of SFH8 clusters (described in **Fig 4D and 4E**), we tested whether full-length SFH8 undergoes LLPS. SFH8$^{IDR}$ was predicted as an inducer of LLPS through the catGranule algorithm, while the corresponding region of a close SFH8 homolog, SFH6 (SFH6$^{IDR}$), was predicted to exhibit a reduced propensity to undergo LLPS (**Fig 5C**); we verified this prediction in *N. benthamiana* where SFH6 could not form PM-localizing clusters (**Fig 5D**). SFH8$^{IDR}$ sequence composition is distinct from that of animal proteins that undergo phase transitions in the cytoplasm with prion-like domains (PLDs) but show an amino acid distribution like that of the average IDR profile for Arabidopsis (**Fig 5E**) [34]. We further observed that puncta formed by the SFH8$^{IDR}$ failed to colocalize with vesicular markers and endosomes (SNX1, PI3P, and FM4-64), tonoplast (TIP), cellulose synthase complex (CESA6), or mitochondria, in Arabidopsis roots (mitotracker; **S5A and S5B Fig**). Furthermore, SFH8$^{IDR}$ showed LLPS hallmarks such as droplet-like dynamic morphology with frequent fission, fusion, and interconnections (**S5C Fig** and **S5** **Movie**). FRAP analysis of these produced mNeon-tagged SFH8$^{IDR}$ puncta demonstrated a rapid signal recovery ($t_{1/2}$ approximately 10 s, mobile fraction approximately 40%) and sensitivity to 1,6-hexanediol, which blocks in many cases LLPS (**S5D and S5E Fig**) [35,36]; 1,6-hexanediol dissolved SFH8 clusters on the PM but not SFH8 filaments (**S5E Fig**), confirming their solid-like properties.

Since SFH8 clusters at the PM displayed properties akin to condensates, we speculated that they may also form by LLPS much like SFH8$^{IDR}$. In silico prediction, using PLAAC (prion-like amino acid composition) and CIDER (classification of intrinsically disordered ensemble regions) showed that SFH8 can adopt context-specific conformational states with an absolute value of net charge per residue (NCPR) of 0.014, which suggests that is a polyampholyte [32]. This result suggested that the propensity of SFH8 to undergo LLPS may be sensitive to the environment (e.g., lipid species) and that SFH8 may represent an ensemble of conformers.

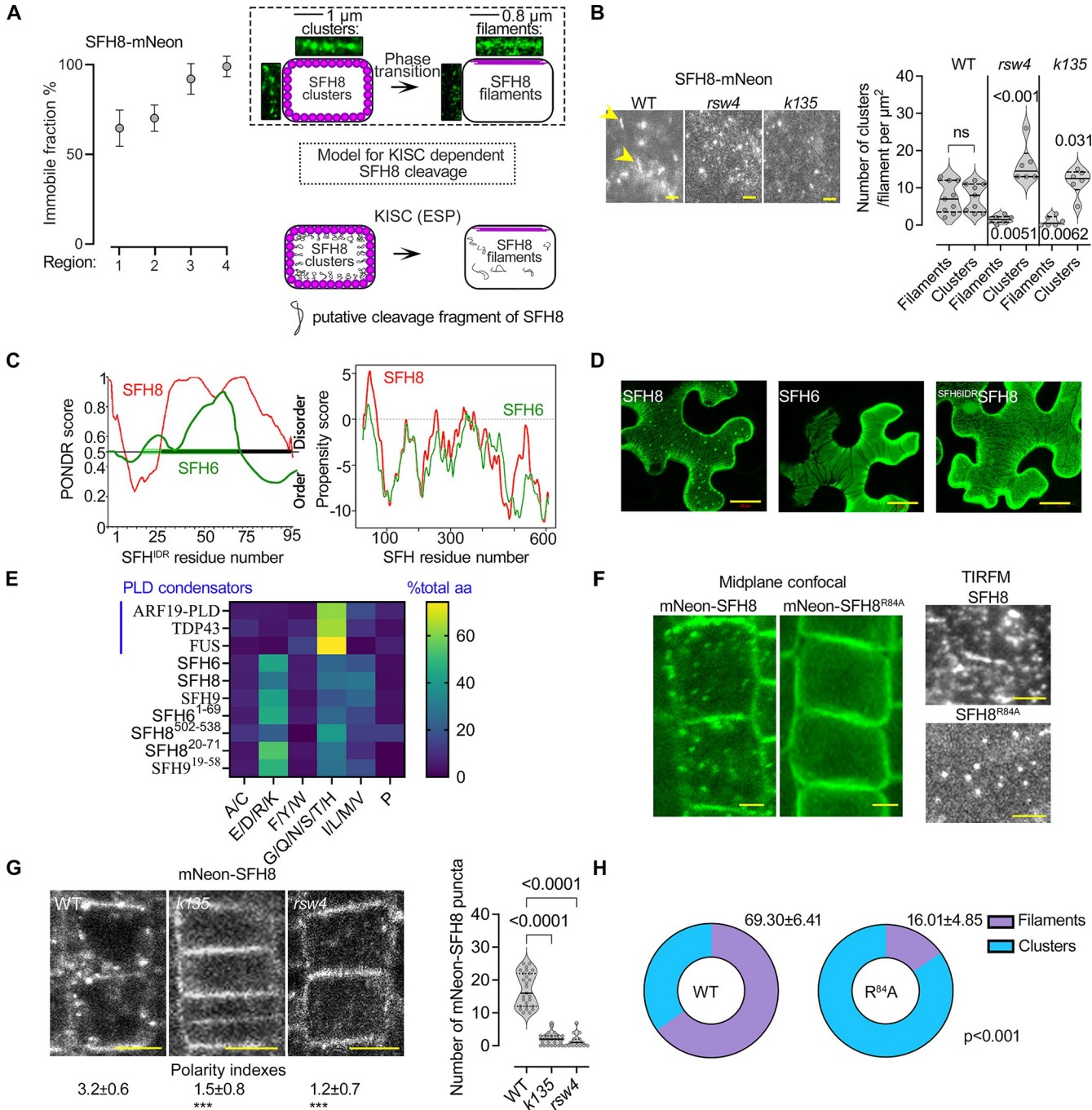

**Fig 5. KISC abrogation retains the clustered phase of SFH8 through SFH8 N-terminus.** (**A**) FRAP analysis of SFH8-mNeon immobile fraction (SFH8pro; 7 DAG, epidermis regions 1–4). Data are means ± SD (*N* = 1, *n* = 10 roots at each point). Right: clusters and filaments in 2 root regions (7 DAG, cortex regions 1 and 4) determined by super-resolution confocal microscopy (midsection; regions 1 and 4, left and right, respectively) and a model showing the SFH8 clusters-to-filaments conversion and its dependence by KISC (relevant to (B)). (**B**) SFH8-mNeon (SFH8pro) localization in WT, *k135*, and *rsw4* root cells (5 DAG, region 3, TIRFM; arrowheads indicate filaments). Scale bar, 0.5 μm. Right: quantification of SFH8-mNeon clusters and filaments in WT, *k135*, and *rsw4* (data are means ± SD, *N* = 3 pooled experiments, *n* = 2–3 roots with 5 fields of view per experiment; *p*-values were calculated by Wilcoxon). Clusters with circularity below 0.5 were defined as filamentous. (**C**) In silico predictions of IDRs by PONDR (left), and phase separation propensity determined by catGRANULE [40] for SFH8 and SFH6 (right). (**D**) Micrographs from *N. benthamiana* leaf epidermis showing the reduced puncta formation in a chimeric protein of SFH6IDR and the C-terminal SFH8 (SDH6IDRSFH8), in the presence of ESP/CyclinD (see also **S3 Fig** for the activation of ESP protein by CyclinD). The experiment was replicated 3 times. Scale bars, 20 μm. (**E**) Comparative analysis of the SFH8 IDR amino acid residue composition. Each amino acid residue is assigned to one of 6 groups on the x-axis, and the fraction of grouped amino acids is shown. For comparison, model "condensators" are shown (ARF19 to FUS). The lengths of the IDRs were determined by the fIDPnn [41]. (**F**) Micrographs (midplane) showing the localization of SFH8R84A (7 DAG, epidermis region 3). Scale bars,

2 μm. Right: persistence of PM SFH8 condensates in *sfh8 SFH8*[R84A] lines (TIRFM, setting as in **Fig 4A**). Scale bars, 0.2 μm. Bottom: the pie graphs show the quantification of mNeon-SFH8 or mNeon-SFH8[R84A] clusters and filaments ($N = 4$ pooled experiments, $n = 122$; $p$-values were calculated by a 2-tailed $t$ test). (**G**) Puncta formation and polarity of mNeon-SFH8 PM signal (SFH8pro) in WT, *k135*, or *rsw4* (7 DAG, epidermis of region 3). Numbers indicate polarity indexes (data are means ± SD, $N = 3$ pooled experiments, $n = 5$–10 cells per experiment; "***": $p < 0.0001$ to WT; $p$-values were calculated by Dunnett). Scale bars, 5 μm. (**H**) Quantifications of mNeon-SFH8 puncta in WT, *k135*, or *rsw4* (7 DAG, epidermis region 3; $N = 3$ pooled experiments, $n = 74$–98 cells per experiment; $p$-values were calculated by ANOVA). Raw data can be found in the Supporting information section (S1 Data). DAG, day after germination; ESP, EXTRA SPINDLE POLES; FRAP, fluorescence recovery after photobleaching; IDR, intrinsically disordered region; KISC, kinesin-separase complex; PLD, prion-like domain; PM, plasma membrane; *rsw4*, *radially swollen 4*; SFH8, SEC FOURTEEN-HOMOLOG8; TIRFM, total internal reflection fluorescence microscopy; WT, wild type.

The highly electronegative field of the PM where SFH8 accumulates could unbalance opposite charges upon IDR removal (IDR NCPR = 0.071) due to repulsive forces, which would support a filamentous structure. Indeed, LLPS of SFH8 relied on the N-terminal IDR, as swapping it with the corresponding region from the SFH6, reduced clustering at the PM and the formation of puncta in the cytoplasm in the presence of ESP (**Fig 5D**, [SFH6IDR]SFH8). Overall, our analyses suggest that SFH8 behaves like an LLPS polyampholyte at the PM with negatively charged lipids (e.g., phosphatidylinositols (PIs)), likely buffering repulsive charges that restrict condensation.

To further examine LLPS of SFH8, we established an in vitro LLPS assay with fluorescently labelled proteins using thiol-reactive maleimide dyes (see **Materials and methods**). Under conditions that promote phase separation (**S6A Fig** for protein purification), recombinant GST-tagged SFH8 or the uncleavable variant SFH8[R84A] formed condensates at relatively high concentrations (5 μM), while SFH8[ΔIDR] (for delta IDR, i.e., SFH8 without the IDR) formed filament-like assemblies in good agreement with the in vivo situation (**S6B Fig**). Consistent with the in vivo data, the SFH8[ΔIDR] filaments showed no recovery after photobleaching and reduced circularity compared to SFH8 condensates (**S6C Fig**).

As mentioned above, phase separation at the PM could be affected by the interfacing of the condensate with lipids. Hence, as the above tests of bulk phase separation could be less relevant to SFH8, we established a system to test SFH8 phase separation on membranes. We used SUPER templates (supported lipid bilayers with excess membrane reservoir) that contain low-tension membranes surrounding a silicon bead [37]. GST-SFH8 formed large droplets on SUPER templates containing PI lipids (i.e., PI(4,5)P2, as the yeast Sec14 binds on these lipids), at lower concentrations compared to the bulk-phase experiments (**S6D Fig**; 0.1 μM versus $\geq$5 μM in the bulk phase). This result suggested that membranes promote LLPS of SFH8. It is worth noting that, consistent with our data, the threshold concentration for LLPS in 2D systems like the PM can be an order of magnitude lower than in the 3D bulk phase (for example, [38]). By contrast, SFH8[ΔIDR] did not show similar behaviour in this setting and formed oligomers (within 1 h) in native polyacrylamide gel electrophoresis; this behaviour could be also observed for SFH8 in the presence of KIN7.3 and ESP, as ESP cleaves the IDR converting SFH8 to SFH8[ΔIDR] (**S6E Fig**). Hence, as suggested above, PIs may neutralize electrostatic repulsions via counterion-mediated charge neutralization along SFH8, as suggested for other proteins, thereby mediating LLPS [39].

As the N-terminal IDR drives the phase behavior of SFH8, we speculated that an uncleavable variant of SFH8 would fail to undergo a liquid-to-solid transition (i.e., cluster-to-filament). Indeed, in lines expressing the uncleavable mNeon-SFH8[R84A] in *sfh8*, cells lacked cytoplasmic fluorescent puncta, while SFH8[R84A] was apolar and did not convert to filaments as observed for SFH8 in KISC mutants (**Fig 5B and 5F-5H**). As expected, the mNeon-SFH8 fluorescent protein produced by *SFH8pro:mNeon-SFH8* lines showed higher FRAP rates on the PM, as expected (due to cleavage and liquidity of clusters), unlike the corresponding C-terminally tagged SFH8-mNeon (**S7 Fig**). We suggest that SFH8 clusters exhibit LLPS and that SFH8

releases 2 proteolytic "proteoforms": C-terminal SFH8$^{\Delta IDR}$ (converted to solid-like filaments) and the N-terminal SFH8$^{IDR}$ (cytoplasmic liquid-like puncta).

## SFH8 phase separation enables delivery of some polar proteins

SFH8$^{IDR}$ may act as an entropic bristle through random movements around its attachment point on lipids, which could, in theory, exclude access of other proteins to PM regions where uncleaved SFH8 resides [42]. This property would also reduce the probability of full-length SFH8 undergoing filamentous transition due to inter- or intramolecular stereochemical hindrance imposed by the IDR [42]. We thus aimed to decipher the significance of SFH8 phase transition at polar domains. As a relevant readout here, we used PIN2 because KISC plays a role in PIN2 delivery [18], but this choice is not implying a strict link between SFH8/KISC to auxin signalling. We observed that the PIN2-GFP (or α-PIN2 by immunohistochemistry) signal is lower by about 50% at the PM of *sfh8* or KISC mutants (**Fig 6A and 6B**), suggesting that SFH8/KISC are required for PIN2 delivery, stability, and/or maintenance on polar domains. Notably, PIN2 accumulated in endosome-like structures in *sfh8*, while KISC or *sfh8* mutants showed a slightly reduced PIN2 polar delivery mainly in the cortex (**Fig 6B and 6C**). Furthermore, the uncleavable variant of SFH8$^{R84A}$ could not rescue the PIN2 defects of *sfh8* (**Fig 6C**). We also observed increased localization to endosomes and a reduced delivery or maintenance for PIN1 on PM in *sfh8* (likely not for other PINs), but not for nonpolar proteins (the H$^+$-ATPase 1 [AHA1] or PLASMA MEMBRANE INTRINSIC PROTEIN 2a [PIP2a]), discounting a general role for SFH8 in exocytosis (**S8A and S8B Fig**). These results suggest that cleavage of SFH8 and, thus, its conversion to filaments is required for the establishment of some polar PM domains.

Next, we asked whether SFH8 promotes the delivery of polar proteins or their maintenance to the PM. To this end, we used the drug brefeldin A (BFA) to induce intracellular agglomerates of PIN2 in so-called BFA bodies (aggregate of *trans*-Golgi network [TGN] and Golgi). We calculated the endocytosis rate of PIN2 to BFA bodies and the delivery rate from PIN2-positive BFA bodies back to the PM after BFA washout [43]. We further validated BFA experiments with FRAP to measure the rate of PIN2 delivery at the PM. Both assays confirmed that PIN2 delivery to the PM is compromised in *sfh8* and KISC mutants, while PIN2 endocytosis was not, as PIN2-positive BFA bodies were produced at the same rate in the wild type and the *sfh8* mutant; these effects were independent of de novo PIN2 synthesis, as short cycloheximide treatments did not affect delivery rates or dissolution of the PIN2 endosomes in *sfh8* (**Figs 6D-6F**, **S8A and S8B**). Furthermore, SFH8 did not significantly colocalize with clathrin clusters at the PM and *sfh8* did not show defects in endocytosis traced by the FM4-64 or the peptide PEP1, which is internalized by clathrin-mediated endocytosis (**S9 Fig**) [44]. These results likely exclude the possibility that PIN2 removal from the PM is due to increased endocytosis in *sfh8*. Similarly, endocytosis was not affected in KISC mutants [28].

To address the mechanism by which SFH8 might affect PIN2 delivery, we checked SFH8 and PIN2 localization dynamics at the PM. In region 4, SFH8-mScarlet (and KIN7.3), but not mScarlet-SFH8 clusters, colocalized with apicobasal-localized PIN2 and showed similar polarity (**Fig 7A–7C**; PCC approximately 0.9). On the contrary, PIN2-GFP and mScarlet-SFH8 PM puncta showed an anticorrelation of localization, excluding each other in regions 1 and 2, as observed in a super-resolution setting (**Fig 7D**; insets). Thus, SFH8$^{IDR}$ properties in SFH8 clusters may reduce the delivery of proteins like PIN2 at the PM in regions 1 and 2. To address whether the entropic bristle effect is responsible for this exclusion, we evaluated whether the formation of SFH8$^{\Delta IDR}$ and the transition to filaments might permit delivery of PIN2, which would likely be manifested as increased SFH8$^{\Delta IDR}$ (filaments) proximity to PIN2 (as the 2

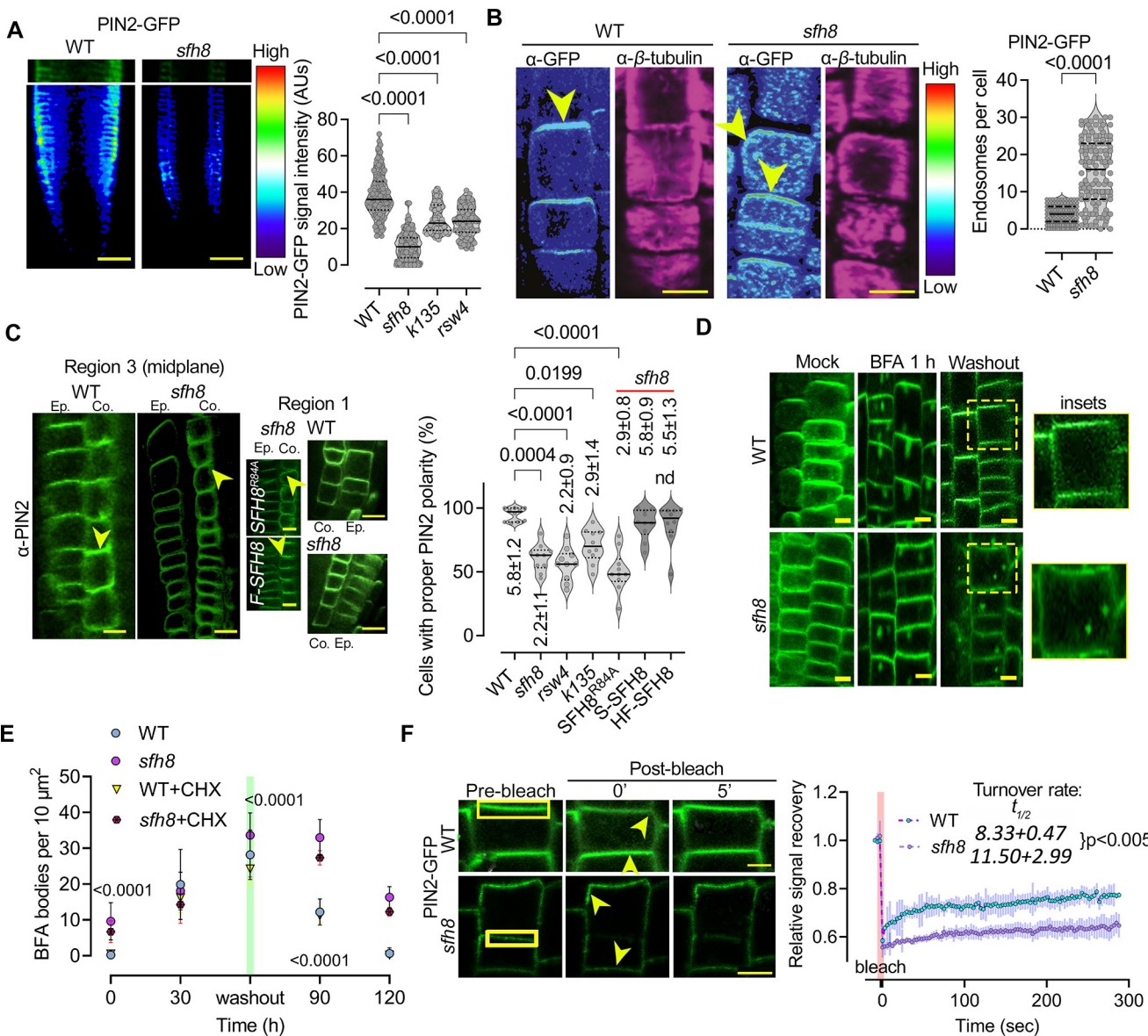

**Fig 6. SFH8 can affect PIN2 dynamics at the PM.** (**A**) Micrographs showing PIN2-GFP signal intensity (colour-coded as spectrum intensity) in WT and *sfh8* and quantification (right) of PIN2-GFP on PM of WT, *sfh8*, *k135*, and *rsw4* (7 DAG, region 3; data are means ± SD, $N = 2$ pooled experiments; $n = 5$–10 cells per experiment; *p*-values were calculated by multiple comparisons Dunnett). Scale bars, 50 μm. (**B**) PIN2 localization (α-PIN2) in WT and *sfh8* (colour-coded as in (A); a-*β*-tubulin staining was used to show focal plane; 5 DAG, region 3). Arrowheads indicate PIN2 accumulation maximum. Note in *sfh8*, the slight polarity offset and the high number of PIN2-positive endosome-like structures. The *sfh8* signal intensity has been adjusted to normalize signal intensity between *sfh8* and WT. Scale bars, 10 μm. Right: quantification of endosomes above the confocal diffraction limit (approximately 200 nm) in WT and *sfh8* under normal conditions (data are means ± SD, $N = 4$ pooled experiments; $n = 25$–34 cells per experiment, 5 DAG, region 3; *p*-value was calculated by ordinary ANOVA). (**C**) PIN2 localization (α-PIN2; 7 DAG, midsection epidermis and cortex region 3) in WT, *sfh8*, and SFH8^R84A *sfh8* (brightness has been adjusted here in *sfh8* and *sfh8* SFH8^R84A), or HF-SFH8 ("F"). Yellow arrowheads denote PIN2 polarity. Right: quantification of cells with proper PIN2 polarity in cortex of WT, *sfh8* (expressing also SFH8^R84A, mScarlet-SFH8 ("S") or HF-SFH8), *k135*, and *rsw4* (data are means ± SD, $N = 10$ pooled experiments, $n = 8$–10 cells per experiment; "*": $p < 0.0001$ to WT; 1-way ANOVA, for the number of cells: $N = 4$, $n = 118$, Kruskal–Wallis). Scale bars, 5 μm. (**D**) PIN2-GFP localization in WT and *sfh8* treated with 50 μm BFA for 1 h and after BFA washout for 30 min (7 DAG, epidermis and cortex region 3). The experiment was replicated 3 times. Scale bars, 4 μm. (**E**) Quantification of BFA bodies (50 μm BFA for 1 h agglomerates ± CHX) in WT and *sfh8*. CHX was added to a final concentration of 30 μM (1 h pretreatment and retained throughout the experiment). Data are means ± SD ($N = 3$ pooled experiments; $n = 5$ fields of view per experiment; *p*-values were calculated by a paired 2-tailed *t* test between WT/SFH8 in the presence of BFA). Scale bars, 5 μm. (**F**) FRAP from polarized PIN2 (7 DAG, epidermis region 3) in WT and *sfh8*. Note the offset of PIN2 polarity (yellow arrowheads) in *sfh8*. The rectangular denotes the bleached ROI. The experiment was replicated twice. Scale bars, 3 μm. Right: quantification of the corresponding PIN2 signal recovery. Data are means ± SD ($N = 2$ pooled experiments, $n = 5$–10 cells per experiment). The red faded band parallel to the y-axis indicates laser iteration time ("bleach"). Numbers next to the genotype, denote recovery half-time ($t_{1/2}$) ± SD (*p*-value was calculated by a paired 2-tailed *t* test). Raw data can be found in the Supporting information section (S1 Data). BFA, brefeldin A; CHX, cycloheximide; DAG, day after germination; FRAP, fluorescence recovery after photobleaching; PIN, PINFORMED; PM, plasma membrane; ROI, region of interest; *rsw4*, *radially swollen 4*; SFH8, SEC FOURTEEN-HOMOLOG8; WT, wild type.

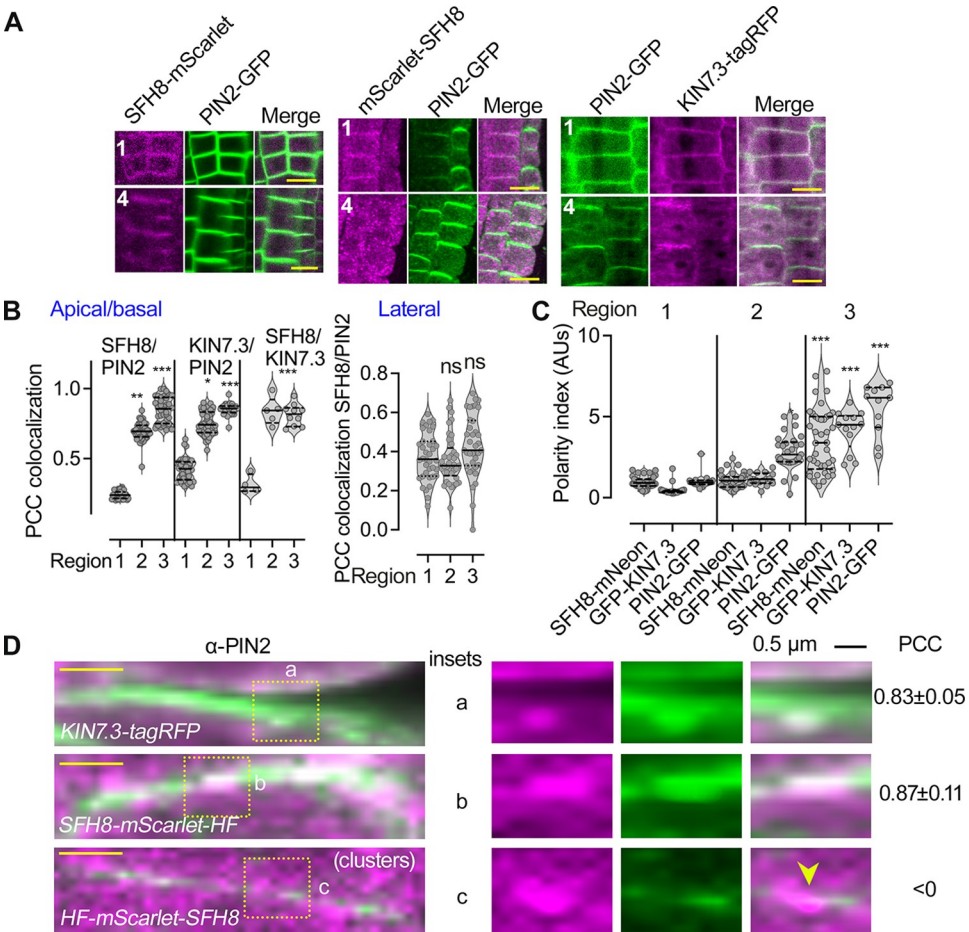

**Fig 7. SFH8 restricts PIN2 delivery when it is uncleaved.** (**A**) Micrographs of tagged with mScarlet uncleaved SFH8 (RPS5apro; region 1) and cleaved SFH8 (converted to SFH8$^{\Delta IDR}$; region 4) that can colocalize with PIN2-GFP for C-terminally tagged SFH8 (left) but not with N-terminally tagged SFH8 (middle; 7 DAG, epidermis regions 1 and 4). Right: micrographs showing the colocalization of HF-KIN7.3-tagRFP (RPS5apro) with PIN2-GFP (7 DAG, epidermis regions 1 and 4). The experiment was replicated 3 times. Scale bars, 5 μm. (**B**) Quantifications of PCC colocalization between SFH8/PIN2, KIN7.3/PIN2, or KIN7.3/SFH8 (apicobasal or lateral domains, for KIN7.3; data are means ± SD, $N = 3$ pooled experiments, $n = 10$–15 cells per experiment; "*": $p < 0.01$, "***": $<0.001$, "****": $<0.0001$, to region 1; $p$-values were calculated by nested 1-way ANOVA). ns, nonsignificant. (**C**) Quantifications of polarity index for SFH8, KIN7.3, and PIN2 (data are means ± SD, $N = 3$ pooled experiments, $n = 4$–10 cells from each root region per experiment; "*": $p < 0.01$, "***": $<0.001$, "****": $<0.0001$, to region 1; $p$-values were calculated by nested 1-way ANOVA). (**D**) Super-resolution micrographs with insets showing details of HF-mScarlet-SFH8 (RPS5apro) cluster/ PIN2 exclusion (7 DAG, epidermis region 3 for the upper 2 micrographs and region 2 for the lower micrograph "clusters"). The experiment was replicated 3 times. Scale bars (left micrographs), 1 μm. Right: PCC values represent colocalization analyses between KIN7.3 or SFH8 with PIN2, while clusters of SFH8 (region 2) showed anticorrelation (denoted by the arrowhead in the inset "c" and low PCC). Data are means ± SD ($N = 3$, $n = 36$–36 measurements on PM per experiment). Raw data can be found in the Supporting information section (S1 Data). DAG, day after germination; PCC, Pearson correlation coefficient; PIN, PINFORMED; SFH8, SEC FOURTEEN-HOMOLOG8.

proteins colocalize). Using a specific antibody against SFH8 (identifying the variable C terminus), we observed that SFH8 localization is not affected in a *pin2* mutant, suggesting that SFH8 affects PIN2 delivery, and not the other way around (**S10A Fig**). To test for interactions between PIN2 and SFH8 in root cells, we refined a quantitative proximity ligation assay (PLA; [45]). PLAs use complementary oligonucleotides fused to antibodies to determine the frequency with which proteins of interest find themselves nearby (**Fig 8A**). We observed positive interactions between PIN2 and SFH8 in regions 3 and 4 (i.e., when SFH8 is in the form of

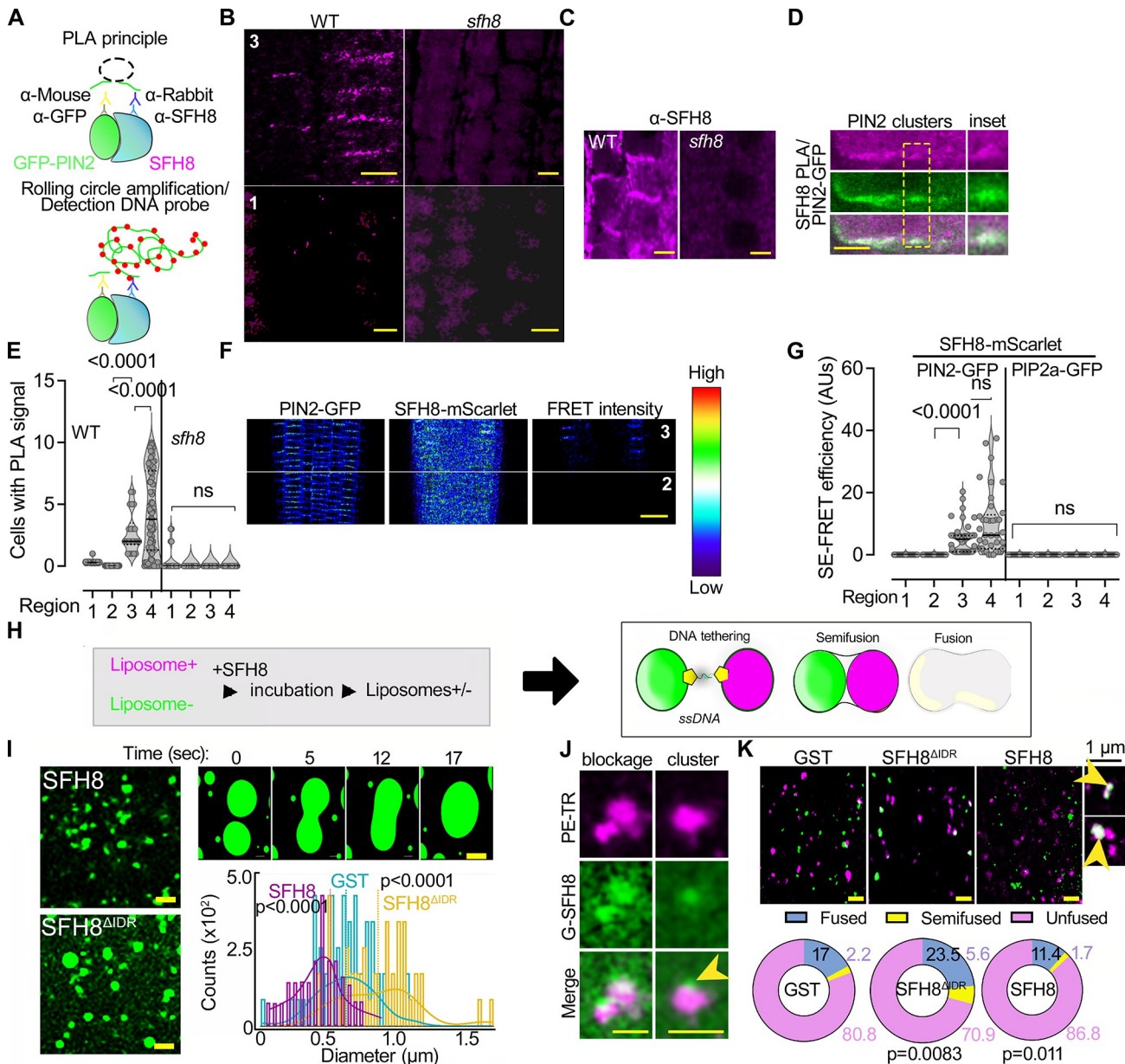

**Fig 8. Filaments of SFH8 promote its interactions and establish more accessible interfaces at the PM.** (**A**) PLA principle. See also [13]. (**B**) PLA-positive signal produced by SFH8/PIN2-GFP interaction (α-SFH8/α-GFP) when SFH8 is in its truncated form (SFH8$^{\Delta IDR}$) in region 3 onwards (7 DAG). Note the lack of PLA when SFH8 is full length. *sfh8* was used as a negative control. The experiment was replicated 3 times. Scale bars, 5 μm. (**C**) α-SFH8 signal localization in WT ("*sfh8*" is a negative control with only background signal). Scale bars, 5 μm. The experiment was replicated 3 times. (**D**) Details of PLA-positive PIN2/SFH8 signal puncta at the PM. Scale bar, 1 μm. (**E**) Quantification of PIN2/SFH8 PLA signals in 4 regions of WT or *sfh8* (data are means ± SD, *N* = 4 pooled experiments, *n* = 10–30 cells per experiment; *p*-values were calculated by a nested 1-way ANOVA). (**F**) SE-FRET efficiency (colour-coded as spectrum intensity) in regions 2 and 3 (5 DAG, epidermis and cortex) between PIN2-GFP and SFH8-mScarlet (RPS5apro). The 2 proteins interact in epidermis and cortex (note the "FRET intensity" micrograph). The spectrum intensity scale is shown next to the micrographs. The experiment was replicated 3 times. Scale bar, 50 μm. (**G**) Quantification of SE-FRET between SFH8/PIN2 and SFH8/PIP2a (data are means ± SD, *N* = 3 pooled experiments, *n* = 12–15 cells per experiment; *p*-values were calculated by a paired 2-tailed *t* test). (**H**) A minimal system to detect the effects of proteins in stereochemical hindrance during fusion, using DNA zippers that bring together liposomes and promote their fusion. If SFH8 would exert stereochemical hindrance (due to the entropic bristle effect), liposome fusion would be blocked. (**I**) DNA zipper assay with GST-SFH8$^{\Delta IDR}$ or -SFH8 (full length; liposomes; lumen was labelled with fluorescein only). The enlarged micrographs (upper right) show a time series of the tethering/fusion of 2 liposomes that converted to GUVs in the presence of SFH8$^{\Delta IDR}$. Scale bar, 2 μm. Bottom right: quantification of corresponding fusion events (data are means ± SD, *N* = 3 pooled experiments, *n* = 80–100 liposomes per experiment; means indicated with vertical lines; *p*-values were calculated by Wilcoxon). (**J**) Super-resolution micrographs showing the fusion blockage by fluorescently labelled SFH8 (G-SFH8) clusters on liposomes (stained with PE-Texas red [magenta]). Arrowhead denotes an SFH8 cluster formed on the LUV liposome (images after deconvolution). The experiment was replicated 3 times. Scale

bars, 2 μm. (**K**) Super-resolution micrographs showing liposome (LUVs) content mixing with lumen stained with fluorescein (green), and lipids stained with PE-Texas red (magenta). Scale bars, 2 μm. Insets (right) show a hemifusion event (upper), and a combination of hemifusion with a tethered LUV (lower inset). The arrowheads show content mixing (pseudo-coloured white). Lower: quantification of the distribution (%) of LUVs in fused, unfused, and hemifused in the presence of recombinant GST, GST-SFH8, or GST-SFH8$^{\Delta IDR}$ (data are means ± SD, $N$ = 3 pooled experiments, $n$ = 32–40 fields of view per experiment; $p$-values were calculated by Dunnett for "fusion" relative to GST). Raw data can be found in the Supporting information section (S1 Data). DAG, day after germination; FRET, Förster resonance energy transfer; GST, glutathione S-transferase; LUV, large unilamellar vesicle; PIN, PINFORMED; PLA, proximity ligation assay; PM, plasma membrane; SE, sensitized emission; SFH8, SEC FOURTEEN-HOMOLOG8; WT, wild type.

SFH8$^{\Delta IDR}$) in epidermis and cortex, using 2 different settings: (i) in roots expressing PIN2-GFP (PLA antibody combination α-GFP/α-SFH8) or (ii) in roots expressing PIN2-GFP and a C-terminally HF-tagged SFH8 (PLA antibody combination α-GFP/α-FLAG). We observed significantly lower PLA signals between PIN2 and SFH8 in the N-terminally tagged HF-SFH8 lines (α-PIN2/α-FLAG) as the full-length SFH8 showed anticorrelation of localization with PIN2. We detected no PLA signal for (i) SFH8 and a PM aquaporin (PIP2a-GFP; α-GFP/α-SFH8), (ii) in the *sfh8* mutant (α-PIN2/α-SFH8), and (iii) in the vasculature where PIN2 is absent (**Figs 8B–8E and S10B–S10D**). PLA also showed that KIN7.3 interacts with SFH8 at the PM (**S10C Fig**; region 3 onwards), confirming the result from the rBiFC in **Fig 1B**. To follow these results in a live imaging setting, we used Förster resonance energy transfer (FRET) analyses, in which we detected high FRET efficiency for the SFH8-mScarlet/PIN2-GFP pair, indicative of interaction, in epidermis or cortex of regions 3 and 4 (**Fig 8F and 8G**). Collectively, these results suggest that SFH8 filamentous conversion (SFH8$^{\Delta IDR}$) allows the association of proteins like PIN2 with the PM.

We then asked whether the observed anticorrelation between SFH8 cluster signal and PIN2 might indeed imply stereochemical hindrance through the entropic bristle effect imposed by full-length SFH8 that could restrain delivery of proteins such as PIN2. We thus established an in vitro membrane fusion assay as a proxy of stereochemical hindrance at membranes based on cholesterol-modified DNA zippers (lipid-DNA-zippers; **Fig 8H**). DNA zippers promote fusion in the absence of other proteins [46]. To test SFH8 effect on fusion, we used lipid-DNA-zippers assays with low content of labelled phosphatidylethanolamine (PE)-Texas red to decorate the periphery of large unilamellar vesicles (LUVs), which also contained the dye fluorescein (LUVs; 400 nm). Under our super-resolution settings, we resolved 3 events driven by DNA zippers: membrane tethering, hemifusion (lipid mixing), and fusion resulting in the unification of the lipid bilayer and the intermixing of the volumes (**Fig 8H**). As SFH8$^{\Delta IDR}$ converted to filaments in a few minutes, to ascertain that observed effects would not be due to differential binding of SFH8$^{\Delta IDR}$ on LUVs (or other surfaces) due to its rapid conversion to filaments, we used quartz crystal microbalance with dissipation (QCM-D) to monitor SFH8 or SFH8$^{\Delta IDR}$ binding on LUVs in real time (**S10E and S10F Fig**; method details in the figure legend). QCM-D can reveal the interaction dynamics between lipids/proteins and/or the sensor surface, translating differential binding of proteins in an observable real-time response. The detection is based on measurements that depend on real-time mass changes [5], i.e., how much protein would bind on the sensor surface. In QCM-D, both SFH8 and SFH8$^{\Delta IDR}$ show only basal affinities toward LUVs (and to KIN7.3 tail), thus excluding differential binding as a potential driver of the changes in fusion dynamics. In the SFH8$^{\Delta IDR}$ samples, the average diameter of liposomes was approximately 1 μm, in contrast to SFH8 (approximately 0.5 μm), which was below that of free GST samples (approximately 0.7 μm) (**Fig 8I**). Content mixing analyses showed that almost 30% of the SFH8$^{\Delta IDR}$ samples show semi-fuse/fused LUVs (approximately 2-fold lower for full-length SFH8), while fusion/hemifusion events with SFH8 were even less than those with GST (**Fig 8J and 8K**). This result suggests that the N-terminal SDH8$^{IDR}$ when on SFH8 exerts an entropic bristle effect blocking the delivery of proteins, while SFH8$^{\Delta IDR}$ allows or even promotes this delivery.

## SFH8 cleavage by KISC mediates developmental robustness

Next, we asked whether the KISC-SFH8 link has biological meaning. At the seedling stage, the *sfh8* phenotypes resembled those of KISC mutants showing both reduced root growth, gravitropism (compared to *pin2*), and slower response to gravistimulation; these phenotypes were rescued by tagged SFH8 (**Figs 9A–9C and S11A–S11F**; note the similarity between *sfh8-1* and *sfh8-2*). We did not observe additive phenotypes of seedlings with KISC/*SFH8* mutant combinations (**Fig 9D**; *rsw4 sfh8* and *k135 sfh8*), suggesting functional convergence between KISC and SFH8. Adult mutants also showed a shorter stature, decreased branching, and smaller cotyledons, or leaves (**S11A–S11C Fig**). Yet, *sfh8* produced shorter siliques and exhibited a more severe adult phenotype than the one reported for the KISC mutants (**S11A–S11C Fig**), suggesting additional functions for SFH8. In *sfh8*, the apical meristem length was approximately 2-fold smaller than in the wild type, likely due to lower mitotic activity as defined by the Cell Cycle Tracking system (**Fig 9E and 9F**; [47]). Although SFH8 is a SEC14-like protein and would, therefore, be expected to be involved in lipid homeostasis, its loss of function (or of KISC) did not affect lipid levels at the PM (**S11D Fig**), suggesting that *sfh8* phenotypes did not relate to perturbations in lipid homeostasis. Noteworthy, all the SFH8 constructs used rescued the *sfh8* seedling phenotype (**S11E and S11F Fig**).

We further examined whether the phase transitions of SFH8 (cluster-to-filament) are biologically significant. The uncleavable SFH8 variant (SFH8$^{R84A}$), which cannot form filaments, failed to rescue *sfh8*, and the same was observed for SFH8$^{6KtoA}$ (**S12A and S12B Fig**). On the other hand, deletion of the IDR (SFH8$^{\Delta IDR}$) also led to a significant loss of SFH8 polarity in roots, a lack of root developmental robustness, and only partial *sfh8* rescue (**S12C and S12D Fig**), suggesting that the initial clustering of SFH8 is functionally important. Furthermore, SFH8$^{\Delta IDR}$ only partially rescued *rsw4* and the $^{SFH6IDR}$SFH8 (showing little condensation) showed a moderate rescue of *sfh8* (**S12D and S12E Fig**), highlighting the importance of SFH8 liquid-to-solid transitions.

## Discussion

Here, we identify a condensate undergoing phase transitions on membranes. Collectively, the mechanism of these transitions consists of (i) SFH8 LLPS at the PM mediated by an IDR, (ii) SFH8 interaction with KISC, and (iii) proteolytic cleavage of SFH8 by KISC, followed by a phase transition that allows interactions with polar proteins. This module contributes to developmental robustness by regulating PM domains. Intriguingly, SFH8 phase transition is induced by the highly conserved protease ESP. So far, ESP targets have mainly been linked to functions in dividing cells. Apart from expanding the targets of ESP, this module also uncovers a novel way to regulate LLPS via proteolytic processing. We further speculate that the released N-terminal SFH8$^{IDR}$ retains features of the LLPS SFH8 state even after its cleavage, suggesting a structural memory for condensates. In this direction, further work will reveal whether proteolysis is a general regulator of phase transitions.

Condensation is crucial for polarity establishment; yet, our work might appear counterintuitive, as it starts challenging (or extending) these models by showing that LLPS may simply be a mechanism for reducing polarized secretion. In synapses, for example, condensation exerts an opposite effect to that described here [1]. SFH8 liquid condensates prelude shifts in material properties (from LLPS to likely more solid oligomeric filaments) that promote a functionality switch for SFH8: from blocker to fusion enhancer. Likewise, Dynamin-related proteins (DRPs) oligomerize to remodel membranes [48], while coronaviruses (e.g., SARS-CoV-2) co-opt host proteases for structural reconfiguration that prime activity of SPIKE resulting in the fusion between the virus and its host cell [49,50]. Upon IDR removal, SFH8$^{\Delta IDR}$ was stabilized,

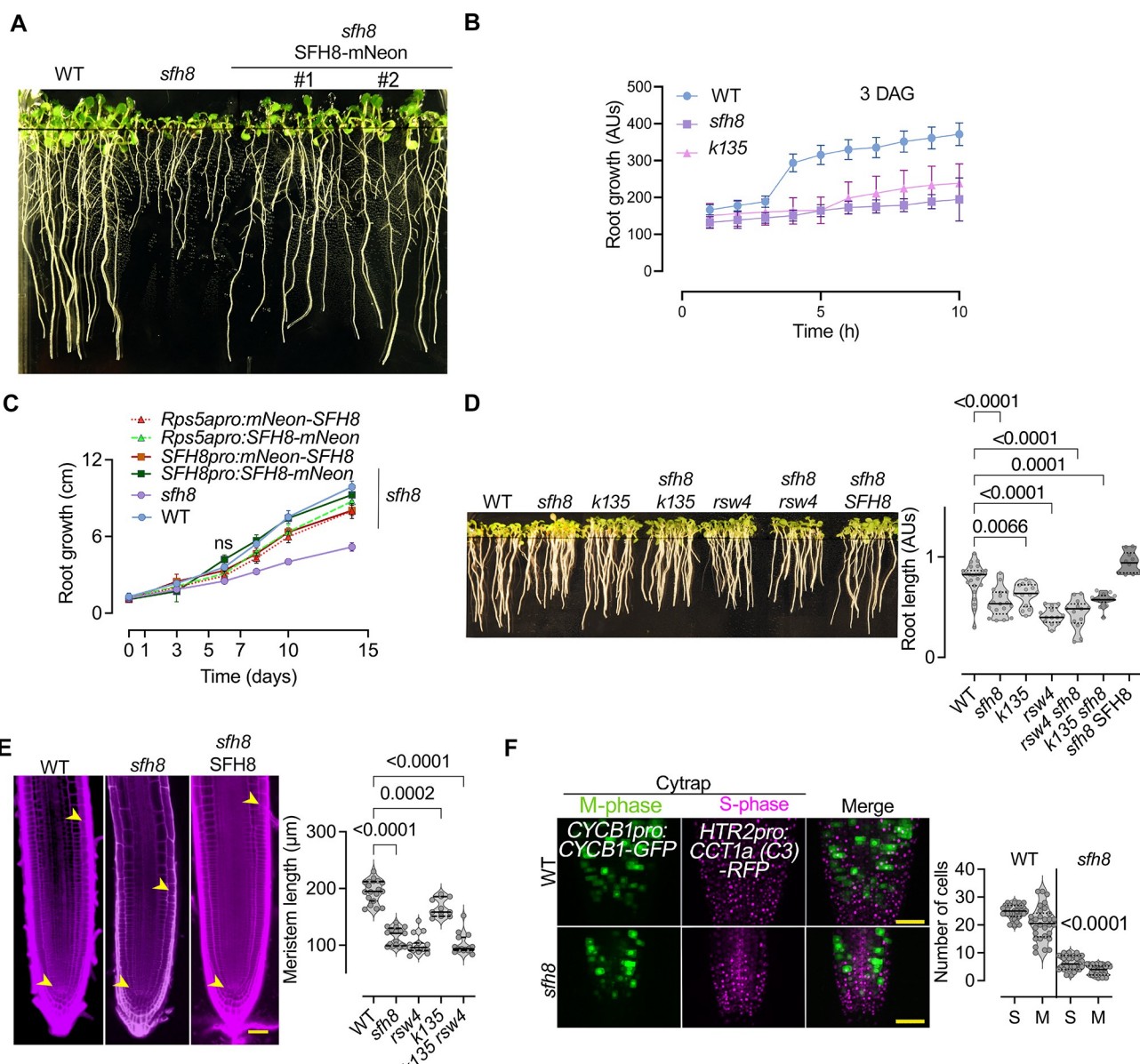

**Fig 9. SFH8 modulates development.** (**A**) Phenotypes of WT, *sfh8-1* (*sfh8* onwards), and *sfh8* rescued seedlings expressing SFH8-mNeon (SFH8pro; 10 DAG). (**B**) Kinematic root growth in the order of hours quantified using "SPIRO" (see also **Materials and methods**; 3 DAG = time 0; data are means ± SD, *N* = 5 pooled experiments, *n* = 8–10 roots per experiment). (**C**) Root growth rate (0–14 DAG) of WT, *sfh8*, *k135*, and rescued *sfh8* expressing (with RPS5apro or SFH8pro, N- or C-terminally tagged with mNeon; data are means ± SD, *N* = 3 pooled experiments, *n* = 8–10 roots per experiment; "*": *p* < 0.01, "***": <0.001, "****": <0.0001, WT vs. *sfh8*; *p*-values were calculated by a *t* test). (**D**) Phenotypes and quantifications of root length of WT, *sfh8*, *k135*, *rsw4*, *rsw4 sfh8* and *k135 sfh8*, and the rescued *sfh8 SFH8* (*SFH8pro:SFH8-mNeon*; data are means ± SD, *N* = 3 pooled experiments, *n* = 7–8 roots per experiment; *p*-values were calculated by a paired *t* test). (**E**) Micrographs of root meristems WT, *sfh8*, and of the rescued *sfh8 SFH8* (5 DAG; red signal: stained cell walls with propidium iodide). Right: quantifications of WT, *sfh8*, or KISC mutants' meristem sizes (data are means ± SD, *N* = 3 pooled experiments, *n* = 19; ordinary ANOVA). The arrowheads indicate the meristem (from QC to the "first elongating cell" showing >50% increase of size along the proximodistal axis). Scale bars, 50 μm. (**F**) Cytrap marker expression in WT and *sfh8* (7 DAG), tracking S and M phases of the cell cycle. Scale bars, 50 μm. Right: corresponding quantifications (data are means ± SD, *N* = 3 pooled experiments, *n* = 9–11 roots per experiment; *p*-values were calculated by a paired *t* test). Raw data can be found in the Supporting information section (S1 Data). AUs, arbitrary units; DAG, day after germination; KISC, kinesin-separase complex; rsw4, radially swollen 4; SFH8, SEC FOURTEEN-HOMOLOG8; WT, wild type.

likely due to reduced intramolecular stereochemical repulsion and charge attenuation, which allowed filamentous state conversion and fusion at polar domains. The stabilization effect of SFH8 filaments may have to do with a steric barrier imposed by the increased size of oligomeric filaments, which would be more resistant to endocytosis. This proposition is in accordance with recent findings suggesting that glycosylation presents a steric barrier for endocytosis by increasing the size of proteins [51]. Overall, this mechanism establishes an example of LLPS with importance for PM micropatterning that transcends to polarized patterns.

Furthermore, the dimensionality reduction caused by the PM binding of SFH8 promotes condensation. Cluster formation through LLPS could have a direct effect on PM properties. Condensates can potentiate lipid clustering [52], and SFH8 could in turn affect lipid clustering with important roles in signalling during development. We believe that these functions may also be relevant during stress, given the important roles of lipids in stress signalling [53]. Furthermore, the conversion of SFH8$^{IDR}$ that has microscale entropic bristle–like properties in mesoscale condensates in the cytoplasm (the SFH8$^{IDR}$-bodies) with large diameter suggests that SFH8$^{IDR}$ can restrict interactions. At this level, these properties of SFH8 could exclude certain proteins from binding to the PM and can likely establish sites for vesicle exclusion. We reconciled this proposition in vitro by showing that the removal of SFH8$^{IDR}$ by ESP locally promoted attraction and fusion by alleviating steric hindrance. Alternatively, SFH8 clusters may engulf diffraction-limited vesicles and promote their fusion upon the removal of the IDR. Because SFH8 is a SEC14-like protein, it may also render lipids vulnerable to enzymatic modifications [54], regulating local lipid environments at the microscale or nanoscale. These lipid modifications could thus promote vesicular fusion. SFH8$^{\Delta IDR}$ may also reduce the energy barrier required for fusion through an increase in fluidity by its filamentous structure, as has been shown for MTs [55], or the reduction of the entropic bristle effect. We provide a model for SFH8 functions in **Fig 10**.

Addressing further how KISC/SFH8 functions is an important priority for our future research. The details of structural modifications for SFH8 especially upon the removal of the SFH8$^{IDR}$ need further exploration. Intriguingly, as aforementioned, the IDR of the SFH8 is conserved throughout evolution (**S1** and **S2 Files**), suggesting that proteins with similar features and functions should exist in other eukaryotes. Other pertinent questions are, 'How do SFH8 clusters form in the first place, and what is the function (if any) of the cytoplasmic "SFH8$^{IDR}$-bodies" condensate?' Our work thus provides insights relevant to condensates interfacing with membranes.

## Materials and methods

### Arabidopsis backgrounds and ecotypes

All the plant lines used in this study were in the Arabidopsis (*Arabidopsis thaliana*) Columbia-0 (Col-0) accession except the ones as indicated individually, and a detailed description can be found in Materials Table. Primers used for genotyping, RT-qPCR, and cloning can be found in **S1 Table**. The following mutants and transgenic lines used in this study were described previously: *rsw4* mutant [56], *k135* [18], *35Spro:mCherry-MAP4$^{MBD}$* [18], *35Spro: smRS-GFP-TUB6*, *35Spro:smRS-GFP-TUA6* (Nottingham Arabidopsis Stock Center-NASC; N6550), *35Spro:GFP-TUB9* (NASC; M84706), *PIN1pro:PIN1-GFP* [57], *PIN2pro:PIN2-EGFP* [58]; *PIN3pro:PIN3-EGFP*, *PIN4pro:PIN4-EGFP* and *PIN7pro:PIN7-EGFP* [59], Cytrap: *HTR2pro:CDT1a(C3)-RFP / CYCB1pro:CYCB1-GFP* [47]. *SNX1pro:SNX1-mRFP* [60], *TIP1-pro:TIP1-GFP* [61], *35Spro:GFP-PIP2a*, and *35Spro:GFP-AHA1* [62], and PI3P marker [63]. In the Wassilewskija background was the *CLC2pro:CLC2-EGFP* [64]. The following lines were

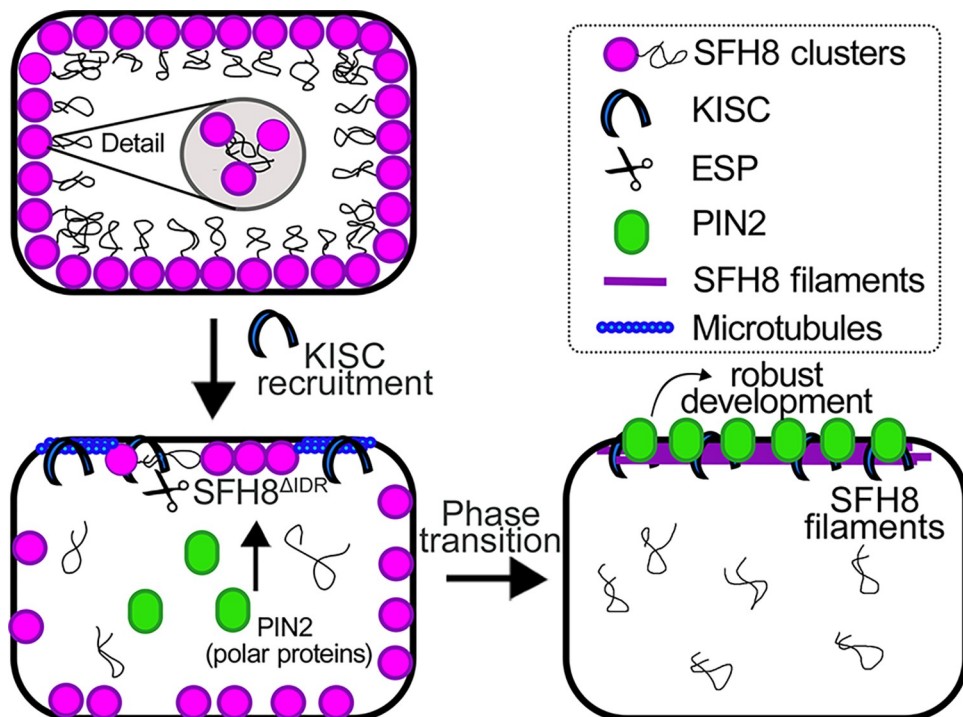

**Fig 10. Model for SFH8 functions during development.** Proposed model for the role of SFH8 in root development. SFH8 (full-length) is recruited on the plasma membrane in a nonpolar manner and forms LLPS clusters at the PM (cell 1). These clusters seem to block the delivery of polar proteins to the PM (e.g., PIN2). Later, KISC is recruited on SFH8, and KISC proteolytic part (ESP) cleaves the N-terminal part of SFH8, which results in the production of SFH8$^{\Delta IDR}$ (SFH8 lacking the N-terminal IDR; cell 2). The released IDR floats in the cytoplasm in the form of LLPS puncta. Microtubules may direct the KISC at the PM; however, microtubules/KISC do not remain associated at the PM. Later, SFH8$^{\Delta IDR}$ ages to a filamentous form with clear polarity at the PM and interacts with other polar proteins, like, for example, PIN2 (cell 3). This phase transition of SFH8 and its polar localization are essential components of robust root development. The released IDR persists in the cytoplasm and has an unknown role. ESP, EXTRA SPINDLE POLES; IDR, intrinsically disordered region; KISC, kinesin-separase complex; LLPS, liquid–liquid phase separation; PIN, PINFORMED; PM, plasma membrane; SFH8, SEC FOURTEEN-HOMOLOG8.

ordered from GABI or NASC: *sfh8-1*: GABI_55IF03, *sfh8-2*: SALK_006862. Arabidopsis plants were transformed according to [65] using *Agrobacterium tumefaciens* strain GV3101. In all experiments, plants from T1 (colocalization experiments), T2, or T3 (for physiology experiments) generations were used. The Arabidopsis fluorescence marker lines were crossed with *sfh8-1* and corresponding SFH8 transgenic lines to avoid the gene expression level differences caused by positional effects; F1 (colocalization experiments) and F2/F3 were used in experiments.

## Plant growth conditions

Arabidopsis seedlings were surface sterilized and germinated on ½ strength Murashige and Skoog (MS) agar medium with sucrose, under long-day conditions (16-h light/8-h dark) and were harvested, and treated, or examined as indicated in the context of each experiment. In all experiments involving the use of mutants or pharmacological treatments, the medium was supplemented with 1% (w/v) sucrose, unless stated otherwise. Arabidopsis plants/lines for crosses, phenotyping of the aboveground part, and seed harvesting were grown on soil in a plant chamber at 22˚C/19˚C (or 28 to 30˚C for restrictive temperature treatments in the case of *rsw4*), 14-h/10-h light/dark cycle, and light intensity 150 μE m$^{-2}$ s$^{-1}$. *N. benthamiana* plants

were grown in Aralab or Percival cabinets at 22˚C (16-h/8-h light/dark cycle, and light intensity 150 µE m$^{-2}$ s$^{-1}$.

## Phenotypic analyses and drug treatments

For quantification of phenotypes, seeds were surface sterilized, plated on MS medium, and seedlings were grown vertically. Customized Smart Plate Imaging Robot (SPIRO) imaging was done with 15-min intervals of fully automated imaging acquisition (https://www.alyonaminina.org/spiro), in a growth cabinet (Aralab). Arabidopsis Col-0 was used as the wild-type control. To define root length, images were captured of the plates using a Leica DM6000 with a motorized stage and computationally compiled together. Root length or size was determined using Image J/Fiji (National Institute of Health). For 1,6-hexanediol treatments, a 10% (v/v) aqueous solution was used. XVE-driven expression was activated by transferring seedlings on ½ MS plates containing various estradiol (in ethanol; light-tight tube aliquots) concentrations, which were determined experimentally in trial experiments (2 to 100 µM). Cycloheximide, APM, latrunculin B, and BFA treatments were done in liquid ½ MS, as described previously [13,18,28].

## Root gravitropism assays

To observe root gravitropism, seedlings were grown vertically on plates (½ strength MS agar medium under long-day conditions (16-h light/8-h dark) for 12 DAG. The root tip angle change was measured using Fiji.

## Cloning and plasmids

Primer sequences used for amplicons are listed in **S1 Table**. Cloning was performed either by Gateway, restriction enzyme digestion, or In-fusion (Takara). The following constructs were produced in (i) pENTR vectors were generated via BP reaction with pDONR/Zeo (Invitrogen) and PCR products: SFH genes (amplicons from RT-PCR using cDNA from 1-week-old seedlings), truncations of SFH8 (amplicons from pDONR/Zeo-SFH8 or SFH6 were used as a template), mutations of SFH8 by mutagenesis PCR using pDONR/Zeo-SFH8. The KIN7.3 and truncations were previously described [18]; (ii) In pGWB601: *SFH8pro*:*mNeon-gSFH8* (g, for genomic), *SFH8pro*:*gSFH8-mNeon*, *RPS5apro*:*mNeon-gSFH8*, *RPS5apro*:*gSFH8-mNeon*, *KIN7.3pro*:*mNeon-gSFH8*, *KIN7.3pro*:*gSFH8-mNeon*, *RPS5apro*:*HF-mScarlet-gSFH8-mNeon*, *RPS5apro*:*HF-gSFH8*, *RPS5apro*:*gSFH8-HF*, and *KIN7.3pro*:*CFP-cKin7.3-mNeon*. The pGWB601 empty vector was used as a backbone and was cut open by XhoI and SacI. Then, the vectors were assembled through 4 or 5 fragments using In-fusion cloning (amplicons from Arabidopsis genomic DNA, template for *mNeon*, *mScarlet*, and *KIN7.3pro*:*CFP-cKIN7.3)*; (iii) pGBKT7/pGADT7-gateway-compatible [66] for Y2H: LR reaction with the pENTR vectors of KIN7.3 and truncations, SFH8, SFH8 truncations, and mutations; (iv) rBiFC-gateway-compatible system [23]: KIN7.3 (pDONR/P3P2-KIN7.3) and SFH8 (pDONR/P1P4-SFH8); (v) RPS5apro gateway compatible dual tagged vectors: modified pGWB517 and 560 empty vectors were used as a backbone, cut open by HindIII and XbaI, and then the vectors assembled by 2 fragments In-fusion cloning (amplicons from Arabidopsis genomic DNA, 1.6 kb *RPS5a* promoter, and template for *FLAG* and *sGFP*). The *KIN7.3* (pDONR/Zeo-KIN7.3) and *SFH8* (pDONR/Zeo-SFH8) clones were used to generate the FRET pair and the cleavage biosensor, respectively; (vi) pGAT4 and pDEST15 gateway compatible vectors for His- and GST-protein production in *Escherichia coli*: full-length or truncations of *KIN7.3* and *SFH8* pDONR/Zeo vectors were used; (vii) inducible constructs under the KIN7.3 promoter (XVE): vector was cut with PmeI and MluI and then the vectors assembled by 2 fragment In-fusion cloning

(amplicons from Arabidopsis genomic DNA 1.7 kb promoter of *KIN7.3* and part of LexA); inserts from pENTR-GFP-ESP and pENTR-HA-KIN7.3 tail were introduced by LR reaction from the corresponding pDONR/Zeo vectors; (viii) the pDR gateway compatible vector for yeast temperature-sensitive complementation: LR reaction with pDONR/Zeo-SFH8 and the indicated truncations (in pDONR/Zeo). The SYN4 cloning has been described in [27].

## Yeast two-hybrid screening and paired interactions

The genotype of the strain Y2HGold is *MATa, trp1-901, leu2-3, 112, ura3-52, his3-200, gal4Δ, gal80Δ, LYS2::GAL1UAS-Gal1TATA-His3, GAL2UAS-Gal2TATA-Ade2, URA3::MEL1UAS-Mel1TATA-AUR1-C MEL1*. The genotype of the strain Y187 is *MATα, ura3-52, his3-200, ade2-101, trp1-901, leu2-3, 112, gal4Δ, gal80Δ, URA3::GAL1UAS-GAL1TATA-lacZ*. Transformed yeast cells were incubated at 30°C until $OD_{600}$ = 0.8 in a minimal medium (SD) lacking the amino acid tryptophan. To confirm the expression of the baits, the total protein (10 µg) was extracted using alkaline lysis and subjected to immunoblot. Fusion proteins were detected with α-Myc monoclonal antibodies (Roche, Stockholm, Sweden). The following constructs were used: pBKGT7-cESP domain A (DomA; [18]; pBKGT7-cESP Domain C (DomC); pBKGT7-cKIN7.3; pBKGT7-cKIN7.3motor (m); pBKGT7-cKIN7.3tail (t) [18]; The absence of self-activation was verified by a transformation of the baits alone to select on minimal medium (SD) lacking the amino acids leucine, histidine, and adenine. The baits were transformed into the strain Y2HGold and mated with the Universal *Arabidopsis* cDNA Library (Clontech) in Y187. For pairwise Y2H assays, the Gateway-compatible pGADT7 vector [66] and the yeast Y187 were used, including the following constructs: pGADKT7-SCC2 [27]; pGADKT7-cKIN7.3-tail; pGADKT7-cSFH8; pGADKT7-cSFH8-SEC14 domain (SD); pGADKT7-cSFH8-Nodulin; pGADKT7-cSFH8-Nodulin[6KtoA].

## Evolutionary relationships of taxa and sequences analyses

The evolutionary history of SFH8 was inferred using the Neighbor-Joining method [67]. The optimal tree is shown. The percentage of replicate trees in which the associated taxa clustered together in the bootstrap test (1,000 replicates) is shown next to the branches [68]. The evolutionary distances were computed using the Poisson correction method [69] and are in the units of the number of amino acid substitutions per site. This analysis involved 201 amino acid sequences. All positions with less than 70% site coverage were eliminated, i.e., fewer than 30% alignment gaps, missing data, and ambiguous bases were allowed at any position (partial deletion option). There was a total of 611 positions in the final dataset. Evolutionary analyses were conducted in MEGA X [70].

## Recombinant protein production and purification from *E. coli*

The pGAT4/PDEST15 constructs were transformed in BL21 (DE3) Rosetta or BL21 (DE3) Rosetta II *E. coli* cells. Bacterial cultures were grown in 800 mL of LB supplemented with 100 mg L$^{-1}$ of ampicillin and 25 mg L$^{-1}$ of chloramphenicol. Protein production was induced at $OD_{600}$ = 0.5 with 0.05 to 1 mM IPTG (isopropyl ß-D-1-thiogalactopyranoside). After 3 h, the cells were harvested by centrifugation at 2,500*g* for 20 min at room temperature (RT) and frozen overnight at −80°C. Preparation of his-tagged recombinant proteins was performed according to manufacturer instructions (Qiagen). Preparation of GST-tagged recombinant proteins was performed according to manufacturer instructions, using Sepharose beads (GE Healthcare Life Sciences), while the pH of purification was 8.3. Expression levels of proteins were estimated by CBB staining in PAGE or by immunoblots. The proteins were dialyzed overnight in assay buffers (2 L).

## Protein immunopurification

Constructs expressing various forms of KIN7.3, SYN4, SFH8, ESP, or TAP-GFP were infiltrated into *N. benthamiana* leaves. Three to four days later, leaves were ground in liquid nitrogen and resuspended in 10 volumes of buffer A (50 mM Tris–HCl [pH 7.5], 5% [v/v] glycerol, 10% [v/v] Ficoll, 0.1% [v/v] Triton X-100, 300 mM NaCl, 5 mM MgCl2, 1 mM EDTA, 1 mM EGTA, plant-specific protease inhibitor cocktail [Sigma], phosphatase inhibitors [Roche], and 1 mM PMSF) and centrifuged at 14,000*g* for 20 min at 4°C. The supernatant was filtered through 4 layers of Miracloth (Calbiochem). For TAP-ESP capture, samples were mixed with immunoglobulin G beads and incubated at 4°C for 1 h with gentle rotation. Beads were precipitated by centrifugation at 300*g*, washed 3 times with buffer A, and treated for 4 h with PreScission protease (GE Healthcare). The TAP-ESP beads were used directly, or the supernatant was incubated for 30 min with nickel beads and his-ESP was eluted with 250 mM imidazole containing buffer A. Protein was dialyzed against 0.1 M PIPES (pH 6.8), 5 mM EGTA, 2 mM MgCl$_2$, and 20% (v/v) glycerol buffer. Protein levels were estimated by immunoblot.

## Immunoblotting

The samples were pulverized using a liquid N$_2$-cooled mortar and pestle, and the material was transferred to a 1.5-ml or 15-ml tube. Extraction buffer (EB; 50 mM Tris–HCl [pH 7.5], 150 mM NaCl, 10% [v/v] glycerol, 2 mM ethylenediamine tetra acetic acid [EDTA], 5 mM dithiothreitol [DTT], 1 mM phenylmethylsulfonyl fluoride [PMSF], Protease Inhibitor Cocktail [Sigma-Aldrich, P9599], and 0.5% [v/v] IGEPAL CA-630 [Sigma-Aldrich]) was added accordingly. The lysates were precleared by centrifugation at 16,000*g* at 4°C for 15 min, and the supernatant was transferred to a new 1.5-ml tube. This step was repeated 2× and the protein concentration was determined by the RC DC Protein Assay Kit II (Bio-Rad, 5000122). Two times Laemmli buffer was added, and equivalent amounts of protein (approximately 30 μg) were separated by sodium dodecylsulfate polyacrylamide gel electrophoresis (SDS-PAGE; 1.0 mm thick 4% to 12% [w/v] gradient polyacrylamide Criterion Bio-Rad) in 3-(N-Morpholino) propane sulfonic acid (MOPS) buffer (Bio-Rad) at 150 V. Subsequently, proteins were transferred onto polyvinylidene fluoride (PVDF; Bio-Rad) membrane with 0.22-μm pore size. The membrane was blocked with 3% (w/v) BSA fraction V (Thermo Fisher Scientific) in phosphate buffered saline-Tween 20 (PBS-T) for 1 h at RT, followed by incubation with horseradish peroxidase (HRP)-conjugated primary antibody at RT for 2 h (or primary antibody at RT for 2 h and corresponding secondary antibody at RT for 2 h). The following antibodies were used: rabbit α-GST (Sigma, G7781, 1:5,000) mouse α-FLAG-HRP (Sigma-Aldrich, A8592, 1:2,000), rat α-tubulin (Santa Cruz Biotechnology, 1:1,000), rabbit α-GFP (Millipore, AB10145, 1:10,000), α-mouse (Amersham ECL Mouse IgG, HRP-linked whole Ab [from sheep], NA931, 1:10,000), α-rabbit (Amersham ECL Rabbit IgG, HRP-linked whole Ab [from donkey], NA934, 1:10,000), and α-rat (IRDye 800 CW Goat α-Rat IgG [H + L], LI-COR, 925–32219, 1:10,000). Chemiluminescence was detected with the ECL Prime Western Blotting Detection Reagent (Cytiva, GERPN2232) or SuperSignal West Femto Maximum Sensitivity Substrate (Thermo Fisher Scientific, 34094). The bands were visualized using an Odyssey infrared imaging system (LI-COR) or Azure Sapphire Biomolecular Imager.

## SFH8 in vitro cleavage assays

GFP-ESP was extracted from infiltrated *N. benthamiana* leaves as above using GFP-TRAP (CromoTek). GFP-ESP protein lysates were mixed with GST beads carrying GST-SFH8 or the corresponding mutants. The samples were left agitating at 37°C for 1 to 2 h.

## Preparation of liposomes

1,2-dioleoyl-sn-glycero-3-phosphocholine (DOPC), 1,2-dioleoyl-sn-glycero-3-phospho-L-serine (DOPS), 1,2-dioleoyl-sn-glycero-3-phospho-(1'-myo-inositol-4′,5′-bisphosphate) (PI(4,5)$P_2$), and 1,2-dioleoyl-sn-glycero-3-[(N-(5-amino-1-carboxypentyl)iminodiacetic acid) succinyl] (DGS-NTA(Ni)) lipids were purchased from Avanti Polar Lipids (Alabaster, Al, USA). Dipalmitoyl phosphatidylinositol 3-phosphate (PI3P) lipids were from Echelon Biosciences Incorporated (EBI). Lyophilized lipids were dissolved in 1:1 (v/v) chloroform: methanol mixed in glass flasks in the desired amount. The organic solvent was first evaporated under a gentle stream of nitrogen while gently turning the flask to form a thin lipid film onto the wall of the flask. The lipid film was further dried with nitrogen for at least 30 min. Lipid films were hydrated in a buffer containing 10 mM Tris–HCl (pH 7.5) (Merck), and 150 mM NaCl at a final concentration of 2 mg/ml. After vortexing for 30 min, the resulting multilamellar vesicle solutions were extruded 25 times through a polycarbonate membrane with 50 or 200 nm nominal pore diameter (LiposoFast, Avestin) leading to a stock solution of unilamellar vesicles, which were stored at 4°C and used for 5 d at most.

## Protein labelling for LLPS

The dyes used for labelling were Alexa Fluor 647 C2 Maleimide and Alexa Fluor 555 C2 Maleimide (Thermo Fisher Scientific). The dyes were dissolved in water-free DMSO (dimethyl sulfoxide) at a concentration of 1 mM. After the protein isolation with GST-affinity chromatography, the proteins were dialyzed against 20 mM Tris–HCl (pH 7.5) and 150 mM NaCl. Proteins were partially labelled in solution in a ratio of 1 mol protein to 0.1 mol of dye (1:10 molar ratio labelled/unlabeled) for 15 min on ice. The labelled proteins were kept at 4°C and used within the next 3 d.

## Preparation of surfaces for suspended liposomes

Before the experiments, Au-coated AT-cut crystals (QSX-301 Biolin Scientific, Q-Sense, Sweden) were immersed in 2% (v/v) Hellmanex solution and rinsed with double distilled water. The gold sensors were further dried using nitrogen flow and cleaned using UV-ozone cleaner (E511, Ossila, Sheffield, UK) for 30 min. The QCM-D experiments were performed using the Q-Sense E4 (Biolin, Q-Sense, Sweden) instrument and AT-cut quartz disks (5 MHz). All the experiments were performed in a buffer solution under a constant flow rate of 50 μL/min at 25°C. Briefly, the gold surface was equilibrated with buffer, followed by neutravidin adsorption (200 μL, 0.2 mg/mL protein solution). 5′-biotinylated, 3′-cholesterol-modified DNA was then used (200 μL, 0.075 pmol/μL) as a binding anchor to liposomes of different lipid compositions as previously described [71]. All liposome solutions were used at a concentration of 0.2 mg/ml and a final volume of 70 μL. Finally, protein solutions of various concentrations were added at a volume of 500 μL.

## Supported lipid bilayer with DGS-NTA(Ni)

Supported lipid bilayers containing 1% (v/v) DGS-NTA(Ni) were formed on $SiO_2$-coated sensors (Q-Sense QSX 303) upon the addition of 0.05 mg/mL lipid solution. Final frequency and dissipation changes were $\Delta f = -170$ Hz and $\Delta D = 0.13 \times 10^{-6}$. These values are typical of the formation of a homogeneous supported lipid bilayer.

### Liposome DNA-mediated fusion assay and SUPER template setting

To establish DNA-zippers, the concentrations of oligomers-cholesterol were adjusted, and various temperatures (4 to 25˚C) were used to achieve DNA zipper–driven fusion at a percentage lower than approximately 10% (noise). Eventually, all liposomes irrespective of the conditions used, given enough time fused to a percentage of approximately 50%. The liposomes used were unilamellar vesicles (LUVs) of 400 nm diameter, with the following composition: DOPC 71.5%, DOPS 27.5%, PE-Texas Red conjugated 0.5% and PI(4,5)P$_2$ 0.5% (PS was included for SUPER template experiment to provide required membrane flexibility). We also used large LUVs without PE-Texas Red lipid. Instead, they carried the fluorophore fluorescein (cf, 100 mM fluorescein in Tris–HCl [pH 7.5]). For the fusion of the 2 different liposome populations, we incubated each with a DNA primer for 45 min at RT. The sequence of the single-stranded DNA molecules can be found in **S1 Table**. For every liposome, 100 DNA molecules were used [72]. SUPER templates were prepared, as described previously [37]. Silica beads (monodispersed 4.9 μm in diameter; Corpuscular, Cold Spring, NY) were added to a premixed solution of liposomes in LLPS buffer in a total volume of 100 μL in a 1.5-mL clear polypropylene centrifuge tube. Glass coverslips (Fisher Scientific, Pittsburgh, PA) were cleaned with piranha solution (concentration H$_2$SO$_4$/30% H$_2$O$_2$ 4:1, v/v) for 1 h at room temperature and washed extensively with boiling water. The coverslips were allowed to attain RT, stored underwater, and air-dried before use. To coat the coverslips with PEG-silane, piranha-cleaned coverslips were first dried in an oven at 200˚C for 3 h and then treated at RT for 30 min with PEG-silane (2%, [v/v]; Gelest, Morrisville, PA) in acetone. The coverslips were later washed extensively with water and air-dried before use. The samples were observed with a confocal LIGHTNING SP8 module (high speed and resolution of 120 nm) using an observation chamber and an Apochromat 63× objective (NA = 1.4). Images were deconvoluted by Leica's built-in software.

### Phase separation assays

The slides and coverslips used for confocal microscopy were treated overnight with 1 to 2 mg/mL PLL-PEG, after cleaning with 2% (v/v) Hellmanex for approximately 2 h, at RT. The LLPS buffer used was the following: 10 mM HEPES (4-(2-hydroxyethyl)-1-piperazine ethane sulfonic acid), 150 mM NaCl, 0.1 mM EDTA, 2 mM DTT (pH 7.4), with and without 10% (w/v) polyethylene glycol (PEG 3000) as a crowding agent. The amount of the added labelled protein was 1 to 2 μM for 3 h at RT. Some of our samples contained LUVs consisting of DOPC 0.5%: DOPS 99%: DOPE-Texas Red 0.5%. Phase separation was imaged via confocal microscopy.

### Whole-mount proximity ligation assay (PLA) and immunocytochemistry

Four- to five-day-old seedlings were fixed and permeabilized as described [18]. The α-SFH8 was raised against the unique peptide sequence 517-GNAIELGSNGEGVKEECRPPSPVPDL-TET-545 in rabbits using standard immunogenic procedures. Primary antibody combination 1:200 for α-GFP mouse [Sigma-Aldrich, SAB2702197] and 1:100 for α-SFH8 rabbit, 1:100 for α-mNeon mouse [Chromotek, 32F6], and 1:100 for α-SFH8 rabbit or 1:200 for α-FLAG mouse [Sigma-Aldrich, F1804] and 1:200 for α-GFP rabbit [Millipore, AB10145] were used for overnight incubation at 4˚C. Roots were then washed with MT-stabilizing buffer (MTSB: 50 mM PIPES, 5 mM EGTA, 2 mM MgSO$_4$, 0.1% [v/v] Triton X-100) and incubated at 37˚C for 3 h either with α-mouse plus and α-rabbit minus for PLA assay (Sigma-Aldrich, Duolink). PLA samples were then washed with MTSB and incubated for 3 h at 37˚C with ligase solution. Roots were then washed 2× with buffer A (Sigma-Aldrich, Duolink) and treated for 4 h at 37˚C in a polymerase solution containing fluorescent nucleotides as described (Sigma-Aldrich, Duolink). Samples were then washed 2× with buffer B (Sigma-Aldrich, Duolink), with 1% (v/v)

buffer B for another 5 min, and then the specimens were mounted in Vectashield medium (Vector Laboratories). Immunocytochemistry was done as described previously [18]. The primary antibodies used were rabbit α-SFH8 (diluted 1:500), rat α-tubulin YL1/2 (1:200; Santa Cruz Biotechnology), mouse α-FLAG (1:250), and sheep α-PIN2 (1:500). Specimens were washed 3 times for 90 min in PBS-T and incubated overnight with donkey α-sheep conjugated Alexa Fluor 488, goat α-mouse tetramethylrhodamine isothiocyanate (TRITC), α-rat TRITC, and α-rabbit fluorescein isothiocyanate-conjugated (FITC) secondary antibodies diluted 1:200 to 250. After washing 3× in PBST, specimens were mounted in Vectashield (Vector Laboratories) medium.

## Transient assays in *Nicotiana benthamiana*

For transient assays, *A. tumefaciens* strain GV3101 was used. Infiltrations were done as described previously [73]. During treatments, plants were kept in defined growth conditions in an Aralab growth chamber.

## Ratiometric bifluorescence complementation and protoplast transformation

The rBiFC assay was done in 5 DAG Arabidopsis root protoplasts via polyethylene glycol transformation as described previously [74]. After a 16-h incubation following transformation, protoplasts were observed with a Zeiss LSM780 laser scanning confocal microscope using 40×/ 1.2 W C-Apochromat in multitrack channel mode. Excitation wavelengths and emission filters were 514 nm/band-pass 530 to 550 nm for YFP, and 561 nm/band-pass 600 to 630 nm for RFP.

## Quantification of fluorescent intensity, FRAP, and FRET

Fluorescence was measured as a mean integrated density in regions of interest (ROIs) with the subtraction of the background (a proximal region that was unbleached and had less signal intensity than the signal of the ROI region). FRAP mode of Zeiss 780 ZEN software was set up for the acquisition of 3 pre-bleach images, 1 bleach scan, and 96 post-bleach scans (or more). Bleaching was performed using 488, 514, and 561 nm laser lines at 100% transmittance and 20 to 40 iterations depending on the region and the axial resolution (iterations increased in deeper tissues to compensate for the increased light scattering). In FRAP, the width of the bleached ROI was set at 2 to 10 μm. Pre- and post-bleach scans were at minimum possible laser power (0.8% transmittance) for the 458 nm or 514 nm (4.7%) and 5% for 561 nm; 512 × 512 8-bit pixel format; pinhole of 181 μm (>2 Airy units) and zoom factor of 2.0. The background values were subtracted from the fluorescence recovery values, and the resulting values were normalized by the first post-bleach time point and divided by the maximum point set maximum intensity as 1. The objective used was a plan-apochromat 20× with NA = 0.8 M27 (Zeiss). For FRET, the sensitized emission (SE) module of the SP8 Leica confocal microscope was used, with standard modules and internal calibration. Fluorescence was detected using a water- or oil-corrected 40× objective. For SE-FRET, the correction factors β, α, γ, and δ were calculated with the donor- and acceptor-only reference samples, as described for TMK1–AHA1 interaction [75].

## TIRFM imaging and tracking analyses

TIRF microscopy images were acquired using MetaMorph software on an Olympus IX-81 microscope. The system was maintained at 37°C during imaging. A DV2 image splitter (MAG

Biosystems) was used to separate GFP and RFP emission signals. Time-lapse movies were obtained at 100-ms intervals. For MSD analysis, 30-s-long movies with 100-ms intervals and 200-ms exposure were used. Particle tracking was limited to the amount of time that PM remained in a single focal plane; the median track length was 2,000 frames, corresponding to 3.5 s of imaging. The tracking of particles was performed with the Mosaic suite or Nano-TrackJ/TrackMate of Fiji, using the following typical parameters: radius 3 of fluorescence intensity, a link range of 1, cutoff of 0.1%, and a maximum displacement of 8 pixels, assuming Brownian dynamics.

## Super-resolution imaging

Confocal SP8 Leica confocal microscope LIGHTNING module was used for super-resolution on-the-fly imaging. Confocal images were obtained when indicated at maximum scanning speed (40 frames per second) using a 63× water immersion objective with a theoretical x/y-axial resolution of 120 nm upon deconvolution. Post-acquisition, images were deconvoluted using the LIGHTNING algorithm, and water correction was set in the algorithm as the mounting medium. Imaging was done at RT in an inverted microscope setting.

## LC–MS/MS lipids and proteins analyses and overlay assays of lipids

Lipids were extracted from roots after inactivating the tissue with boiling water or from microsomes [76]. Lipid classes were purified by solid-phase extraction, and phospholipids, glycolipids, diacylglycerol, triacylglycerol, and total fatty acids were measured by quadrupole time of flight mass spectrometry (Q-TOF MS/MS) or gas chromatography–flame ionization detection (GC-FID), respectively, as previously described [77]. For proteomic analyses of FLAG-tagged SFH8 immunoprecipitates, samples were analyzed by LC–MS using Nano LC–MS/MS (Dionex Ultimate 3000 RLSCnano System) interfaced with Eclipse Tribrid mass spectrometer (Thermo Fisher Scientific). Samples were loaded onto a fused silica trap column Acclaim PepMap 100, 75 μm × 2 cm (Thermo Fisher Scientific). After washing for 5 min at 5 μL/min with 0.1% (v/v) trifluoroacetic acid (TFA), the trap column was brought in line with an analytical column (Nanoease MZ peptide BEH C18, 130 A, 1.7 μm, 75 μm × 250 mm, Waters) for LC–MS/MS. Peptides were fractionated at 300 nL/min using a segmented linear gradient 4% to 15% B in 30 min (where A: 0.2% formic acid, and B: 0.16% formic acid, 80% acetonitrile), 15% to 25% B in 40 min, 25% to 50%B in 44 min, and 50% to 90% B in 11 min. Solution B then returned at 4% for 5 min for the next run. The scan sequence began with an MS1 spectrum (Orbitrap analysis, resolution 120,000, scan range from M/Z 350 to 1,600, automatic gain control (AGC) target 1E6, maximum injection time 100 ms). The top S (3 589 s) and dynamic exclusion of 60 s were used for the selection of Parent ions for MS/MS. Parent masses were isolated in the quadrupole with an isolation window of 1.4 591 m/z, AGC target 1E5, and fragmented with higher-energy collisional dissociation with a normalized collision energy of 30%. The fragments were scanned in Orbitrap with a resolution of 30,000. The MS/MS scan range was determined by the charge state of the parent ion, but the lower limit was set at 100 amu.

## TAMRA-PEP1 and FM4-64 internalization assays

The peptide PEP1 (ATKVKAKQRGKEKVSSGRPGQHN) was labelled with 5′-801 carboxy tetramethylrhodamine at the N-terminus (TAMRA-PEP1) with an HPLC purity of 95.24% and molecular weight of 2,905.24 (EZBiolab). The peptide was dissolved in water to obtain 1 mM peptide stocks. Further dilutions were done with ½ MS medium. Five-day-old seedlings were dipped into 1 mL ½ MS medium containing 100 nM TAMRA-PEP1 for 10 s, washed 5 times, and kept in 24-well plates with ½ MS medium for 40 min. Epidermal cells at the

meristematic zone were imaged with a Zeiss LSM710 inverted laser scanning confocal equipped with 40×/1.2 W C-Apochromat M27 objective at a zoom factor of 3.5. TAMRA-PEP1 was excited at 559 nm, and fluorescence emission was captured between 570 and 670 nm, GFP was excited at 488 nm, and fluorescence emission was captured between 500 and 540 nm. The FM4-64 experiments were as described in [28]. In brief pulse labelling with FM4-64 (2 μM FM464, Molecular Probes; made from a 2-mM stock in DMSO) was done for 5 min (time 0) and then analyzed for 18 min at RT in Zeiss LSM 780 with excitation at 488 nm and fluorescence emission captured between 540 and 670 nm. After the pulse with FM4-64, the roots were washed 2 times in an ice-cold MS medium.

## Quantification and statistical analyses

Statistical analysis was performed in R studio (R-project.org) or GraphPad Prism (version 9.2.0). Each data set was tested for normal distribution (Gaussian) by the Shapiro–Wilk, D' Agostino–Pearson, Anderson–Darling, and Kolmogorov–Smirnov normality tests. Lognormal versus Gaussian distributions were also evaluated (not used though herein). For tests involving pairwise comparisons, Student $t$ test, or Wilcoxon tests (or as indicated) were mainly used to define whether differences were statistically significant. The significance threshold was set at $p < 0.05$ (significance claim-level alpha), and the exact values are shown in graphs ($p$-values <0.0001 are indicated as such). For tests involving multiple comparisons, 1-way ANOVA, or the Kruskal–Wallis test (nonparametric analogue of ANOVA) was used followed by Dunnet or Dunn multiple comparison tests to define whether differences were statistically significant (or as indicated). Graphs were generated by using Microsoft Excel, R, or GraphPad Prism 9.2.0. Details of the statistical tests applied, including the choice of statistical method, and the exact number of "n" is indicated in the corresponding figure legend or directly on the graph. In violin plots, upper and lower dotted lines represent the first and third quantiles, respectively, horizontal lines mark the mean, and edges mark the highest and lowest values. Plots were depicted as truncated or untruncated for aesthetic reasons. "N" corresponds to biological replicate and "n" to technical.

## Image analyses

Image analyses and intensity measurements were done using Fiji v. 1.49 software (rsb.info.nih. gov/ij). The intensity of the fluorescence signal was measured as the Integrated Density in an ROI. PMs of individual cells were selected with the brush tool with a size of 10 pixels as well as the intracellular space with the polygon selection tool. The average intensity of the top 100 highest pixels for both the plasma membrane and the intracellular space was used to obtain a ratio between intracellular and PM fluorescence. The dwell time rate of tagged proteins in FRAP experiments was calculated by the single exponential fit equations as described previously [78]. Colocalization was analyzed using Pearson statistics (Spearman or Manders analyses produced similar results) [79]. Images were prepared by Adobe Photoshop v. 2021 (Adobe). Statistical analyses were performed with JMP v. 9 or 11 (www.jmp.com), GraphPad, or R. Curve fitting was done as we have described in detail for PCs [80]. Time series movies were compressed, corrected, and exported as.avi extension files. The unspecific fluorescence decay was corrected using Fiji v. 1.49 software and default options using the bleaching correction tool. Videos were digitally enhanced with Fiji-implemented filters, correcting noise using the Gaussian blur option and pixel width set to 1.0. Mean intensities in FRAP have been normalized (1 corresponds to pre-bleach signal intensity in arbitrary units).

## Supporting information

**S1 Movie. TIRFM of a diffusive cluster following a directional track.**
(AVI)

**S2 Movie. TIRFM of diffusive clusters following nondirectional tracks.**
(AVI)

**S3 Movie. Nondiffusive clusters with low dwelling times.**
(AVI)

**S4 Movie. Immobile patches of SFH8 atop short filaments resembling a "beads on a string" localization pattern.**
(AVI)

**S5 Movie. SFH8$^{IDR}$ cytoplasmic puncta showing dynamic morphology with frequent splitting, fusion, and interconnections resembling liquid–liquid phase-separated (LLPS) condensates.**
(AVI)

**S1 File. SFH-like proteins alignments.**
(PDF)

**S2 File. IDR region conservation in various SFH-like proteins.**
(TIF)

**S1 Table. Oligos and their sequences used in the study.**
(XLSX)

**S1 Data. Raw measurements used to generate each graph in its respective figure panel.**
(XLSX)

**S1 Raw Images. Uncropped images of western blots described in this study.**
(PDF)

**S1 Text. Descriptions of interactors identified in Y2H.**
(DOCX)

**S1 Fig. KISC polar localization at the PM.** (**A**) Apical/basal/lateral PM domain nomenclature and polarity index determination. (**B**) Micrographs of α-ESP/α-KIN7.3 localizations at similar polar domains (counterstained with α-β-tubulin; region 4, experiment replicated 4 times). Scale bars, 5 μm. (**C**) Micrograph of the ESP-expressing estradiol inducible line (*KIN7.3-pro>XVE>GFP-ESP*; 5 DAG; 20 μM estradiol, 16 h). *KIN7.3* promoter was used, as the *ESP* promoter was not functional. Scale bars, 5 μm. ESP showed polarization only in the distal meristematic region, while in the proximal meristem region, ESP was apolar. Micrographs were obtained in low resolution (512 × 512 with minimal exposure and no averaging), due to photobleaching and low expression levels of ESP (experiment replicated more than 10 times with variable expression). Scale bar, 50 μm. (**D**) Digitally zoomed-in confocal micrographs (deconvoluted, 5 DAG) from root tip cells of the *KIN7.3pro>XVEpro>GFP-ESP*. Merist., proximal meristematic cell (note the diffused cytoplasmic signal). Ep., epidermis; Co., cortex; Cell pl., cell plate; Cap, lateral root cap. Note the MT-binding of ESP in the lateral root cap cell where MTs are highly bundled (as also described in [18,28]). (**E**) Micrographs of roots expressing *RPS5apro:KIN7.3-mNeon* in the *k135* background at the indicated regions (5 DAG). Right: micrographs from lines expressing *RPS5apro:KIN7.3-tagRFP* in the *k135* background (5 DAG). CP, cell plate; PH, phragmoplast; na, not taken from the micrograph on the left. The

experiment was replicated 4 times. Scale bars, 20 μm. Raw data can be found in the Supporting information section (S1 Data). DAG, day after germination; Ep., epidermis; ESP, EXTRA SPINDLE POLES; KISC, kinesin-separase complex; MT, microtubule; PM, plasma membrane. (TIF)

**S2 Fig. KISC association with the lipid-transfer protein SFH8 and complementation of *sec14-1^{ts}* yeast mutant by SFH8.** (**A**) Y2H between BD fusion with KIN7.3 (bait) and 5 putative interactors (AT2G21520 (SFH8) PP2CA1, DNAJ, SPC25, and PORcino fused to activation domain (pray); see also **S1 Text** for a description of the KIN7.3 interactors, which were not followed herein). DDO, double dropout (control); QDO, quadruple dropout (selection of interactions). Negative controls: p53 and lamin (Lam). Right: Kin7.3 tail (t, C terminus) or Kin7.3 motor (m, N terminus) interaction with SFH8 lacking the nodulin motif (1–479) or with the nodulin motif (479–673). For the architecture of SFH8, see (C). The experiment was replicated 5 times. (**B**) Micrograph (maximum intensity projection) showing SFH8-GFP localization in *N. benthamiana* leaf epidermis (3 days post infiltration and driven by the 35Spro). Scale bar, 20 μm. The experiment was replicated 20 times. (**C**) SFH8 architecture; numbers indicate aa truncations used throughout the paper. SD1 corresponds to amino acid residues 1–478. IDR, intrinsically disordered region (1–96 aa). The CRAL-TRIO domain binds small lipophilic molecules and is named after the cellular retinaldehyde-binding protein and TRIO guanine exchange factor. Lower: phylogenetic analysis of SFH8. Numbers on branches indicate bootstrap values with a confidence cutoff of 70. (**D**) Budding yeast temperature-sensitive *sec14-1^{ts}* loss-of-function mutant complementation by full-length SFH8 or SD1 (1–479; see (B)). At 37˚C, the complementation was moderate, likely due to the instability of the protein at elevated temperatures (physiological Arabidopsis growth temperature which is between 22–28˚C). Dilution series: 0 (undiluted)–$10^{-4}$. The experiment was replicated 3 times. Raw data can be found in the Supporting information section (S1 Data). BD, binding domain; KISC, kinesin-separase complex; SFH8, SEC FOURTEEN-HOMOLOG8; Y2H, yeast two-hybrid. (TIF)

**S3 Fig. ESP cleavage of the cohesin SYN4 and SFH8 in vitro.** (**A**) Establishment of a positive cleavage control for ESP: the mitotic cohesin SYN4 is an in vivo ESP target (see also relevant **S1 Text** on the use of SYN4). Micrographs of epidermal root cells expressing *SYN4pro*:*CFP-SYN4-YFP* in WT or the temperature-sensitive ESP mutant *rsw4* (at the restrictive temperature of 28˚C for 72 h, 5–7 DAG). Insets show chromosomal bridges in *rsw4* due to the presumptive lack of the mitotic SYN4 kleisin subunit cleavage by ESP (epidermis); the white arrowhead denotes a chromosomal bridge. The experiment was replicated 5 times (with 1–3 bridges evident per root). Scale bars, 40 μm. (**B**) Immunoblots showing the in vitro cleavage of SYN4 on beads purified from leaves of *N. benthamiana* (using myc, *35Spro*:*SYN4-myc*), in the presence of immunopurified from *N. benthamiana* aTAP-ESP (or GFP-aTAP) preactivated by CyclinD (**Materials and methods** for details on purification). Incubation was for 1 h at 37˚C. The experiment was replicated 3 times. (**C**) In vitro GST-SFH8 cleavage by immunopurified from *N. benthamiana* GFP-tagged variants of ESP, ESP^{PD} (protease dead [81], and ESP in the presence of LUVs (made from phosphatidylcholine and containing 10 mol %phospatidylserine). The beads carrying GFP-tagged ESP variants in the ±LUVs were coincubated with GST-SFH8 at 37˚C for 1 h. Note that proteolytically inactive ESP failed to cleave SFH8, while cleavage is sustained in the presence of liposomes (ESP/LUV). The produced GST-SFH8^{IDR} is shown (band below 50 kDa). As the observed compromised full-length SFH8 transfer from the SDS-PAGE to the membrane, the immunoreactive signal increment of the SFH8^{IDR}/SFH8 ratio upon cleavage was not proportional to the full-length SFH8 depletion. Note that the band approximately 50 kDa was detected in all samples and thus could correspond to a nonspecific

cleavage product. Detection of GFP-ESP (WT or PD) is shown at the bottom. The experiment was replicated 4 times. (**D**) GST-SFH8$^{R84A}$ is not cleaved by aTAP-ESP (similar experimental setting as in (B)). Immunoblots are representative of an experiment replicated 3 times. Raw data can be found in the Supporting information section (S1 Data and S1 Raw Images). DAG, day after germination; ESP, EXTRA SPINDLE POLES; GST, glutathione S-transferase; LUV, large unilamellar vesicle; PD, protease dead; *rsw4*, *radially swollen 4*; SFH8, SEC FOURTEEN-HOMOLOG8; WT, wild type.
(TIF)

**S4 Fig. SFH8 liquid-like clusters do not colocalize with MTs or KISC, and the positively charged patch of SFH8 is indispensable for localization and polarity.** (**A**) Micrographs of *KIN7.3pro*:*GFP-KIN7.3*/*35Spro*:*tagRFP-MAP4*$^{MBD}$ expressing lines (5 DAG, epidermis region 3) counterstained with FM4-64, ±APM (10 nM, 1 h). Micrographs are representative of an experiment replicated twice. Scale bars, 12 μm. Right: micrographs of lines coexpressing *KIN7.3pro*:*GFP-KIN7.3*/*35Spro*:*tagRFP-MAP4*$^{MBD}$ at cell contours (5 DAG, epidermis region 3). The box at the top right shows PCC calculated at the regions indicated with the arrowheads, which, in the case of MAP4$^{MBD}$, denote PM-attached MT bundles (data are means ± SD, $N$ = 3, $n$ = 5–10 cells per experiment). The white arrowhead denotes a slightly increased colocalization between GFP-KIN7.3/*tagRFP*-MAP4$^{MBD}$. Scale bar, 1 μm. (**B**) Micrograph showing the partial colocalization of GFP-KIN7.3 filaments with MAP4$^{MBD}$ (in root region 3) ±APM (10 nM, 1 h). Note that KIN7.3 filaments can associate with bundled-MT remnants (reminiscent of clusters; insets with arrowheads). Micrographs are representative of an experiment replicated twice. Lower: micrograph showing SFH8 filaments and their resistance to APM (10 nM, 1 h; region 3); colocalization between tagRFP-MAP4$^{MBD}$/α-β-tubulin signals, confirming that tagRFP-MAP4$^{MBD}$ follows tubulin localization. Note though that overexpression of MAP4 protein induces significant MT bundling. Scale bars, 0.1 μm (for filaments) or 3.4 μm for colocalization (tagRFP-MAP4 $^{MBD}$/α-β-tubulin). Right: quantification of KIN7.3/SFH8 filaments' length at the PM ($N$ = 6 pooled experiments, $n$ = 5–9 cells measuring 10 filaments in each; $p$-value was calculated by Wilcoxon). GFP-KIN7.3 showed more variance, indicative of KIN7.3 association with MT filaments/bundles, as well. Scale bars, 1 μm. (**C**) Micrographs of the line coexpressing *KIN7.3pro*>*XVEpro*>*GFP-ESP*/*RPS5apro*:*mScarlet-SFH8*, after estradiol induction (20 μM, 16–24 h; 5 DAG, regions indicated on micrographs-epidermis, cell surface). Images were obtained at maximum scanning speed (40 frames per second) using super-resolution. The experiment was replicated 3 times. Scale bars, 10 μm. (**D**) Micrographs showing mNeon-SFH8 (SFH8pro) localization ±latrunculin (LatB, 1 μM, 1 h) in lines coexpressing LifeAct-mCherry (Ubi10pro; 5 DAG, epidermis region 3). Scale bars, 5 μm. Right: details of SFH8 filaments that do not colocalize with LifeAct-mCherry. Arrowheads indicate actin filaments. The experiment was replicated twice. Scale bars, 10 μm. (**E**) Micrographs showing the propensity of RFP-tagged SFH8 variants (SFH8 and SFH8$^{6KtoA}$ under the 35Spro) to form clusters in *N. benthamiana*. Insets (right) show details of clusters. The experiment was replicated twice. Scale bars, 20 μm. Lower: Y2H between SFH8/SFH8$^{6KtoA}$ with KIN7.3 tail. KIN7.3 tail was fused to the BD and the 2 SFH8 variants to the AD. DDO, double dropout (growth control); QDO, quadruple dropout (selection medium for interaction). Two dilution series are shown (0 and 0.1). The experiment was replicated 3 times. (**F**) Micrographs showing colocalization analyses of RFP-SFH8, and -SFH8$^{6KtoA}$ with GFP-KIN7.3 (*35Spro*:*GFP-KIN7.3*) in *N. benthamiana* transient expression system (3 days post-infiltration). Note that in cells expressing SFH8$^{6KtoA}$, GFP-KIN7.3 retained a cytoplasmic localization (insets and details on the right). Note also the lack of GFP-KIN7.3 colocalization with RFP-SFH8$^{6KtoA}$ clusters, which suggests that K residues and/or lack of "hyper-clustering" promotes the association of SFH8

with KISC. Micrographs are representative of an experiment replicated 3 times. Scale bars, 50 μm. (**G**) Reduced levels and lack of mNeon-SFH8[6KtoA] polarity (green; *RPS5apro:mNeon-SFH8[6KtoA] sfh8*) in roots counterstained with FM4-64 (magenta; 5 DAG epidermis region 3). Micrographs are representative of an experiment replicated 3 times. Scale bar, 25 μm. Raw data can be found in the Supporting information section (S1 Data). AD, activation domain; BD, binding domain; DAG, day after germination; KISC, kinesin-separase complex; MT, microtubule; PCC, Pearson correlation coefficient; PM, plasma membrane; SFH8, SEC FOUR-TEEN-HOMOLOG8; Y2H, yeast two-hybrid.
(TIF)

**S5 Fig. SFH8[IDR] does not colocalize with membrane markers or the cellulose synthase complex.** (**A**) Micrographs from *RPS5apro:mNeon-SFH8* expressing lines counterstained with FM4-64 (endosomes and PM) or mitotracker (mitochondria; upper panel); lines coexpressing *RPS5apro:mNeon-SFH8* with *35Spro:TIP1-RFP* (tonoplast intrinsic protein 1), or *35Spro:SNX1-RFP* (sorting nexin 1; TGN; lower panel). In some cases, the duration of tracking was limited to the amount of time that particles remained in a single focal plane, as the required acquisition rate did not permit the collection of z-stacks (voxels); the median track length was 35 frames, corresponding to 20 s of imaging. PCC analyses failed to show colocalization of SFH8 cytoplasmic puncta with any of the markers/dyes used. The experiment was replicated 3 times. Scale bars, 2 μm. (**B**) SFH8 does not colocalize with PtdIns(3)P-positive structures in vivo (endosomes and autophagosomes). Micrograph of lines coexpressing mScarlet-SFH8-mNeon (green; *RPS5apro:6xhis-3xFLAG(HF)-mScarlet-SFH8-mNeon*; denoted as "cleavage biosensor"; see also **Fig 3**) with 1xPXp40 (tagged with CFP and expressed under pUBI10). Inset (right) shows a lack of colocalization between PtdIns(3)P and mScarlet-SFH8[IDR]. Scale bar, 4 μm. Right: micrograph of lines coexpressing *RPS5apro:mNeon-SFH8* with Tdt-CESA6 (CELLULOSE SYNTHASE 6 under the 35Spro). The experiment was replicated 3 times for various developmental stages (2–7 DAG), and root tissues (regions 1–4). Scale bar, 4 μm. (**C**) Upper: FRAP signal recovery as a fraction of time in lines expressing *35Spro:GFP-DCP1* (forming cytoplasmic condensates known as processing bodies [82]) or *SFH8pro:mNeon-SFH8*. The ROIs were set on mobile cytoplasmic puncta (at the midsection; 7 DAG, epidermis regions 3–4). The experiment was replicated 3 times. Scale bars, 0.4 μm. Lower: time-lapse imaging (2-s time interval) of mNeon-SFH8 puncta. Arrowheads denote the tubulating (left) or coalescing puncta (right). (**D**) FRAP signal recovery as a fraction of time, of DCP1 and SFH8 (relevant to C). The red faded band parallel to the y-axis indicates laser itera-tion time (bleach). Data are means ± SD (*N* = 2 pooled experiments, *n* = 1 assay). (**E**) Micro-graphs from lines expressing *DCP1pro:DCP1-GFP*, *SFH8pro:mNeon-SFH8* or *FLOT1pro:FLOT1:GFP* treated with 10% (v/v) 1 h 1,6-hexanediol (5 DAG; cell surface and midsections, epidermis regions 2–3). FLOT1 decorates microdomains at the PM [83]. The insets are from the PM surface (TIRFM), showing that 1,6-hexanediol dissolved the SFH8 clusters. The experi-ment was replicated 3 times. Scale bars, 5 μm. Raw data can be found in the Supporting infor-mation section (S1 Data). DAG, day after germination; FRAP, fluorescence recovery after photobleaching; PCC, Pearson correlation coefficient; PM, plasma membrane; PtdIns(3)P, phosphatidylinositol 3-phosphate; ROI, region of interest; SFH8, SEC FOURTEEN-HOMO-LOG8; TGN, *trans*-Golgi network; TIRFM, total internal reflection fluorescence microscopy.
(TIF)

**S6 Fig. Recombinant SFH8 phase separation and polymerization.** (**A**) Examples of purified recombinant proteins from *E. coli* for assays used below visualized on SDS-PAGE (10%). We note that variations in each purification were observed. (**B**) Micrographs of in vitro recombi-nant GST, GST-SFH8, GST-SFH8[ΔIDR], and GST-SFH8[R84A] proteins stained with Alexa638 in

the presence or absence of the crowding agent PEG3000 in LLPS conditions. Note that in the presence of PEG3000, SFH8 and SFH8[R84A] switch to agglomerate-like states, while coincubation of GST-SFH8[ΔIDR] with GST-SFH8 induced the filamentous transition of both proteins. The experiment was replicated 3 times. Right detail (fusion): GST-SFH8 droplet fusion in the same experiment. (**C**) Representative confocal high-speed micrographs showing droplet and aggregate signal recovery in FRAP. Scale bars, 0.2 μm. The experiment was replicated 3 times. Lower: FRAP signal recovery as a fraction of time of GST-SFH8[R84A] droplets and aggregates. The red faded band parallel to the y-axis indicates laser iteration time. A quantification of the circularity of condensates or aggregates is also shown (lower right). Data are means ± SD ($N$ = 5 pooled experiments; $n$ = 1 assay). (**D**) Super-resolution micrographs from a SUPER template experiment show that GST-SFH8 labeled with Alexa638 (0.1 μM of GST-SFH8 in the assay) can form liquid-like droplets on membranes. Furthermore, GST-SFH8 showed an increased propensity to undergo LLPS in the presence of SUPER templates (see arrowhead denoting droplet; some droplets were released in the bulk phase). The experiment was replicated 3 times. Scale bar, 20 μm. (**E**) Native gel electrophoresis and detection by α-GST of SFH8[ΔIDR] and full-length SFH8 showing the time-depended conversion of the 2 proteins to high molecular weight assemblies. Very few agglomerations were observed for GST-SFH8. Note that in the presence of Kin7.3/ESP GST-SFH8 converted faster to high-molecular weight assemblies (GST: negative control). Right immunoblot: detection of Kin7.3 using α-his. Immunoblots are from a single representative experiment replicated twice. Raw data can be found in the Supporting information section (S1 Data and S1 Raw Images). FRAP, fluorescence recovery after photobleaching; GST, glutathione S-transferase; LLPS, liquid–liquid phase separation; SFH8, SEC FOURTEEN-HOMOLOG8.
(TIF)

**S7 Fig. N- and C-terminally tagged SFH8 residence times at the PM.** Microgrpahs showing FRAP signal recovery of the N or C-terminally tagged SFH8 with mNeon on the PM (SFH8pro in *sfh8* background, 5–7 DAG, epidermis region 3), implying increased diffusion (due to cleavage, liquidity, and dwelling time of clusters). The yellow rectangular denotes the ROI that was bleached. The experiment was replicated 3 times. Scale bars, 5 μm. Right: corresponding FRAP signal recovery as a fraction of time. Percentages indicate immobile fractions for N- or C-terminally tagged SFH8. The red faded band parallel to the y-axis indicates laser iteration time (bleach). Data are means ± SD ($N$ = 3 pooled experiments, $n$ = 1 assay). Raw data can be found in the Supporting information section (S1 Data and S1 Raw Images). DAG, day after germination; FRAP, fluorescence recovery after photobleaching; PM, plasma membrane; ROI, region of interest; SFH8, SEC FOURTEEN-HOMOLOG8.
(TIF)

**S8 Fig. SFH8 modulates protein delivery or fusion with polar domains at the PM.** (**A**) FRAP signal recovery as a fraction of time from lines expressing *35Spro:GFP-AHA1* in WT or *sfh8* (upper; inset graph without CHX). The red faded band parallel to the y-axis indicates laser iteration time (bleach). Data are means ± SD ($N$ = 3 pooled experiments, $n$ = 3 assays). nd, no difference. (**B**) Micrographs of lines expressing *35Spro:GFP-AHA1*, or *35Spro:PIP2a-GFP* in WT or *sfh8* (5 DAG, epidermis and cortex). The experiment was replicated twice. Scale bars, 20 μm. (**C**) FRAP signal recovery as a fraction of time from lines expressing PIN2-GFP in WT or *sfh8* in the presence of CHX. The red faded band parallel to the y-axis indicates laser iteration time (bleach). Data are means ± SD ($N$ = 3 pooled experiments, $n$ = 5–8 assays). Lower: PIN2-GFP retention in *sfh8* and *k135* endosomes (30 min CHX treatment; 5 DAG, epidermis and cortex region 3). Note that CHX did not lead to the dissolution of PIN2 endosomes. To normalize PIN2 levels and visualize endosomes in mutants, brightness was increased by 50%.

The experiment was replicated twice. Scale bars, 3 μm. (**D**) Micrographs showing the localization of C-terminally tagged PINs at polar domains expressed in WT, *sfh8*, and *rsw4* under native promoters (24 h at the restrictive temperature 28˚C; 5 DAG). Right: localization details for PIN1; note that in some cases, PIN1-GFP accumulated in endosomes in the *sfh8* and *k135 rsw4*, suggesting that PIN1 delivery was also compromised albeit to a lesser extent than PIN2 (as also discussed in [28] for the *rsw4* mutant). The experiment was replicated 10 times. Scale bars, 20 μm. Note that slight level perturbations at the PM were also observed for PIN3, 4, and 7 in the *sfh8* and *k135 rsw4* (24 h at the restrictive temperature 28˚C; 5 DAG), but due to different patterning of the columella cells in these mutants, these are hard to follow. Raw data can be found in the Supporting information section (S1 Data). CHX, cycloheximide; DAG, day after germination; FRAP, fluorescence recovery after photobleaching; PIN, PINFORMED; PM, plasma membrane; *rsw4*, *radially swollen 4*; SFH8, SEC FOURTEEN-HOMOLOG8; WT, wild type.
(TIF)

**S9 Fig. SFH8 and KISC do not significantly affect bulk endocytosis.** (**A**) Dual-channel TIRFM showing the lack of colocalization between mScarlet-SFH8 (magenta) and CLC2-GFP (green). The experiment was replicated 5 times. Scale bar, 0.2 μm. (**B**) Micrographs showing the lack of colocalization between mNeon-SFH8 (SFH8pro, 5 DAG) and TAMRA-PEP1 (left; stains TGN and PM clusters). (**C** and **D**) Quantification of TAMRA-PEP1 and FM4-64 uptake in WT, *sfh8*, and *k135* (violin plot and rate plot, respectively; Data are means ± SD, $N = 3$ pooled experiments, $n = 18–25$ cells per experiment, *p*-values were calculated by Wilcoxon). Scale bars, 2 μm. Raw data can be found in the Supporting information section (S1 Data). DAG, day after germination; KISC, kinesin-separase complex; PIN, PINFORMED; PM, plasma membrane; SFH8, SEC FOURTEEN-HOMOLOG8; TGN, *trans*-Golgi network; TIRFM, total internal reflection fluorescence microscopy; WT, wild type.
(TIF)

**S10 Fig. Controls for SFH8/PIN2 interactions and liposome fusion assays.** (**A**) Micrographs showing SFH8 localization (α-SFH8) signal in WT, or *pin2* mutant (5 DAG, region 3 cortex), and quantification of SFH8 signal intensity (lower; $N = 3$ pooled experiments, $n = 9$ roots with signal calculations from cortex regions 3–4; *p*-values were calculated by unpaired *t* test). Scale bars, 5 μm. (**B**) Technical controls testing the specificity of the PLA approach in our settings (PIN2/SFH8 PLA assay), showing also that detected interactions are selective and SFH8 is not promiscuously interacting with proteins. SFH8 did not interact with PIP2a (α-GFP/α-SFH8), while SFH8-HF (α-FLAG, RPS5apro) interacted extensively with PIN2 at the PM of region 3 onwards (α-GFP; epidermis and cortex). Note the significantly reduced PLA-positive signal in the N-terminally tagged SFH8, HF-SFH8/PIN2 (PLA signal was observed at the cell plate as well). The *sfh8* mutant was used as a negative control for the PLA (sometimes a nuclear, likely nonspecific signal was observed in negative controls). The experiments were replicated 5 times. Scale bars, 50 μm. (**C**) PLA of KIN7.3-mNeon/SFH8 (α-mNeon/α-SFH8). Note the increased interaction towards region 4 (for example, in the vasculature; upper micrographs). The "no -α-SFH8" corresponds to negative control (bottom). The experiment was replicated 5 times. Scale bars, 50 μm (for "region 3" panel, 10 μm). (**D**) α-FLAG signal in HF-SFH8 (N-terminally tagged) or SFH8-HF (C-terminally tagged) expressing lines (RPS5apro) in *sfh8* (5 DAG). Insets denote the localization at the corresponding region. Lower: α-FLAG detection of HF-SFH8 counterstained with α-PIN2 signal (5 DAG). Note the lack of PIN2 from the vasculature and the presence of both SFH8 and PIN2 at the cortex and epidermis. Note also that FLAG-tagged SFH8 localizes like the native SFH8. The experiment was replicated twice. Scale bars, 50 μm. (**E**) We used the QCM device as a sensitive mass sensor monitoring the frequency

response ($\Delta f$) of the acoustic wave during binding events on the surface of a model membrane [5]. The device surface was first covered with neutravidin (layer 1), which was used to bind specifically a 5′-biotinylated DNA (layer 2). This DNA was also modified at the 3′-end with a cholesterol moiety, further employed to anchor a liposome (layer 3). We prepared 50 nm diameter liposomes with the following lipid composition: DOPC:PI(4,5)P2 (99:1, n/n). Finally, proteins (GST-SFH8 or GST-SFH8$^{\Delta IDR}$) were infused (layer 4) on top of layer 3. Right: real-time curves showing dynamic binding through the frequency response of QCM-D obtained upon infusion of full-length SFH8 or GST-SFH8$^{\Delta IDR}$ proteins on LUVs. Differences in the final amount of the bound 2 protein variants (as this is depicted in the final frequency change observed in each case) is considered insignificant. As of note, SFH8/SFH8$^{\Delta IDR}$ differ in molecular mass and thus SFH8 should give a slightly higher response in $\Delta f$ as shown. The experiment was replicated twice. (**F**) Again, due to the filamentous conversion of GST-SFH8$^{\Delta IDR}$ as described above (see E), we designed a real-time binding assay using, in this case, an SLB as a membrane model and monitored the binding of the 2 protein variants, i.e., GST-SFH8 and GST-SFH8$^{\Delta IDR}$, to KIN7.3; the latter has been attached to the SLB via a his-linker. Right: real-time curves showing dynamic binding through the frequency response of QCM-D obtained upon infusion of full-length SFH8 or GST-SFH8$^{\Delta IDR}$ proteins on supported lipid bilayers carrying hexahistidine-tagged KIN7.3 tail immobilized on the SLB via DGS-NTA(Ni) lipid anchor. Differences observed in the binding of the 2 protein variants to KIN7.3 are very small, suggesting a similar binding mechanism. As of note, SFH8/SFH8$^{\Delta IDR}$ differ in molecular mass, and, thus, SFH8 should give a slightly higher response in $\Delta f$ as shown. The experiment was replicated twice. Raw data can be found in the Supporting information section (S1 Data). DAG, day after germination; LUV, large unilamellar vesicle; PLA, proximity ligation assay; PIN, PINFORMED; PM, plasma membrane; QCM-D, quartz crystal microbalance with dissipation; SFH8, SEC FOURTEEN-HOMOLOG8; SLB, supported lipid bilayer; WT, wild type. (TIF)

**S11 Fig. Aspects of phenotypes of *sfh8* and KISC mutants.** (**A**) Phenotype of adult plants from *sfh8-1* (*sfh8*), and *sfh8-1* SFH8 (*proSFH8:SFH8-mNeon*), which rescues *sfh8-1* (21 and 60 DAG). (**B**) Branch number from adult WT, *sfh8-1*, or *sfh8-1* SFH8 (*N* = 1 representative experiment, *n* = 9 plants). Note that this phenotype is consistent with auxin-related defects. (**C**) Phenotypes of adult plants from WT and *sfh8-1* (upper), and their siliques (lower). Note that SFH8 mutation impacts overall growth and fecundity. Length bar, 0.8 cm. (**D**) Photos of WT, *sfh8-1*, and *sfh8-2*, of 5 DAG seedlings (growth in the presence of 0.5% sucrose). Right: leaf phenotype of WT, *sfh8-1* (*sfh8*), or *sfh8* SFH8. Hereafter, *sfh8-1* is denoted as "*sfh8*," and due to phenotypical resemblance with *sfh8-2* allele is used throughout the paper. (**E**) Photos of WT, *sfh8*, and of *pin1* and *pin2* 10 DAG seedlings. The phenotype of gravitropic defects becomes more evident by increasing growth rate with 1.5% sucrose (compare to *pin1* and *pin2* mutants and to WT). Right: circular plots quantitating moderate gravity perception defects of *sfh8*, *k135*, *rsw4*, the double *rsw4 sfh8*, and *k135 sfh8* mutants grown vertically on plates at 11 DAG. Data are from a single representative experiment replicated 4 times (*N* = 4, *n* = 10 seedlings). (**F**) Quantification of a gravistimulation root tip bending assay, in which vertical plates were tilted by 90˚, and the root curvature rate was determined (*N* = 1 representative experiment, *n* = 9 seedlings). The *sfh8* resembled the response of KISC mutants in gravistimulation, as reported in [28]; also note the resemblance between *pin2* and *sfh8*, in terms of slower gravistimulation response. (**G**) LC–MS/MS lipid species quantitative analysis in WT, *sfh8*, *k135*, *sfh8 SFH8*, and the *rsw4* mutant (5 DAG; treated for 24 h at 28˚C). Data are means ± SD (*N* = 5, *n* = 1 seedling; no significance was revealed using parametric and nonparametric tests). (**H**) Photos of seedlings 10 DAG expressing various fusions of SFH8 in the *sfh8* background. The

experiment was replicated 3 times. The confocal micrographs (bottom left) show localization of KN/KC, RN/RC (5 DAG, epidermis region 3). Scale bars, 3.5 μm. Raw data can be found in the Supporting information section (S1 Data). DAG, day after germination; DGDG, digalactosyldiacylglycerol; KISC, kinesin-separase complex; MGDG, monogalactosyldiacylglycerol; PA, phosphatidic acid; PC, phosphatidylcholine; PE, phosphatidylethanolamine; PG, phosphatidylglycerol; PI, phosphatidylinositol; PIN, PINFORMED; PS, phosphatidylserine; *rsw4*, *radially swollen 4*; SFH8, SEC FOURTEEN-HOMOLOG8; SQDG, sulfoquinovosyl diacylglycerol; WT, wild type.
(TIF)

**S12 Fig. Functional importance of SFH8 cleavage and phase transitions.** (**A**) Photo showing partial *sfh8* rescue by *RPS5a:mNeon-SFH8^{R84A}*. The experiment was replicated 3 times. (**B**) Photo showing partial *sfh8* rescue by *RPS5apro:mNeon-SFH8^{6KtoA}* (10 DAG). Right: corresponding quantifications of root growth (data are means ± SD, $N = 2$ pooled experiments; $n = 5$–$11$ roots per experiment; *p*-values were calculated by ordinary 1-way ANOVA). The experiment was replicated twice. (**C**) Micrographs from lines expressing *RPS5apro:mNeon-SFH8^{ΔIDR}* in *sfh8* showing details of localization with a lack of filaments, polarization, and lack of cytoplasmic puncta formation (7 DAG). Also, note the reduced robustness of PM localization, uneven SFH8 levels, and perturbations of growth realized as reduced growth anisotropy (details; middle right cell denoted by arrowhead). The experiment replicated was 3 times. Scale bar, 40 μm and details, 20 μm. (**D**) Photo showing partial *rsw4* rescue by *RPS5apro:mNeon-SFH8^{ΔIDR}* (10 DAG). The experiment was replicated twice. Right: corresponding quantifications of root length (data are means ± SD, $N = 2$ pooled experiments, $n = 9$ seedlings per experiment; *p*-values were calculated by ordinary 1-way ANOVA). (**E**) Photo showing the partial *sfh8* rescue by *RPS5apro:^{SFH6IDR}SFH8* (10 DAG). Right: corresponding quantification of root length (data are means ± SD, $N = 2$ pooled experiments, $n = 6$–$12$ seedlings per experiment; *p*-values were calculated by ordinary 1-way ANOVA). Raw data can be found in the Supporting information section (S1 Data). DAG, day after germination; PM, plasma membrane; *rsw4*, *radially swollen 4*; SFH8, SEC FOURTEEN-HOMOLOG8.
(TIF)

## Author Contributions

**Conceptualization:** Panagiotis Nikolaou Moschou.

**Data curation:** Chen Liu, Andriani Mentzelopoulou, Fotini Papagavriil, Prashanth Ramachandran, Artemis Perraki, Lucas Claus, Electra Gizeli, Panagiotis Nikolaou Moschou.

**Formal analysis:** Chen Liu, Andriani Mentzelopoulou, Fotini Papagavriil, Prashanth Ramachandran, Artemis Perraki, Lucas Claus, Panagiotis Nikolaou Moschou.

**Funding acquisition:** Panagiotis Nikolaou Moschou.

**Investigation:** Chen Liu, Andriani Mentzelopoulou, Fotini Papagavriil, Prashanth Ramachandran, Artemis Perraki, Lucas Claus, Panagiotis Nikolaou Moschou.

**Methodology:** Fotini Papagavriil, Sebastian Barg, Peter Dörmann, Yvon Jaillais, Eugenia Russinova, Electra Gizeli, Gabriel Schaaf, Panagiotis Nikolaou Moschou.

**Project administration:** Panagiotis Nikolaou Moschou.

**Resources:** Panagiotis Nikolaou Moschou.

**Supervision:** Panagiotis Nikolaou Moschou.

**Validation:** Chen Liu, Andriani Mentzelopoulou, Fotini Papagavriil, Philipp Johnen, Electra Gizeli, Panagiotis Nikolaou Moschou.

**Visualization:** Chen Liu, Andriani Mentzelopoulou, Fotini Papagavriil, Electra Gizeli, Panagiotis Nikolaou Moschou.

**Writing – original draft:** Panagiotis Nikolaou Moschou.

**Writing – review & editing:** Chen Liu, Andriani Mentzelopoulou, Fotini Papagavriil, Prashanth Ramachandran, Artemis Perraki, Lucas Claus, Sebastian Barg, Peter Dörmann, Yvon Jaillais, Philipp Johnen, Eugenia Russinova, Electra Gizeli, Gabriel Schaaf, Panagiotis Nikolaou Moschou.

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
