## [Editor Report · Decision Letter 0]

17 Oct 2022

Dear Dr Moschou, 

Thank you for submitting your manuscript entitled "Phase Transitions of a SEC14-like Condensate at Arabidopsis Plasma Membranes Regulate Root Growth" for consideration as a Research Article by PLOS Biology.

Your manuscript has now been evaluated by the PLOS Biology editorial staff as well as by an academic editor with relevant expertise and I am writing to let you know that we would like to send your submission out for external peer review.

Once your full submission is complete, your paper will undergo a series of checks in preparation for peer review. After your manuscript has passed the checks it will be sent out for review. To provide the metadata for your submission, please Login to Editorial Manager (https://www.editorialmanager.com/pbiology) within two working days, i.e. by Oct 19 2022 11:59PM.

Kind regards,

Ines

--

Ines Alvarez-Garcia, PhD

Senior Editor

PLOS Biology

---

## [Decision Letter · Decision Letter 1]

22 Dec 2022

Dear Dr Moschou,

Thank you for your patience while your manuscript entitled "Phase transitions of a SEC14-like condensate at Arabidopsis plasma membranes regulate root growth" was peer-reviewed at PLOS Biology. Please also accept my apologies for the delay in providing you with our decision. The manuscript has been evaluated by the PLOS Biology editors, an Academic Editor with relevant expertise, and by two independent reviewers. 

The reviews are attached below. As you will see, the reviewers find the conclusions novel and interesting, however they both think that it is so packed with data that it is overwhelming to read and give precise instructions on how to restructure and streamline the paper. After consulting with the Academic Editor, we would like to invite you to submit a revision that thoroughly addresses the reviewers' comments. Please simplify the manuscript following a clear logical flow and, as the reviewers advice, delete any unnecessary information and explain the data well.

Given the extent of revision needed, we cannot make a decision about publication until we have seen the revised manuscript and your response to the reviewers' comments. Your revised manuscript is likely to be sent for further evaluation by all or a subset of the reviewers.

We expect to receive your revised manuscript within 3 monthsdelete unnecessary information. Please email us (plosbiology@plos.org) if you have any questions or concerns, or would like to request an extension. 

**IMPORTANT - SUBMITTING YOUR REVISION**

3. Resubmission Checklist

a) *PLOS Data Policy*

b) *Published Peer Review*

d) *Blurb*

Please also provide a blurb which (if accepted) will be included in our weekly and monthly Electronic Table of Contents, sent out to readers of PLOS Biology, and may be used to promote your article in social media. The blurb should be about 30-40 words long and is subject to editorial changes. It should, without exaggeration, entice people to read your manuscript. It should not be redundant with the title and should not contain acronyms or abbreviations. For examples, view our author guidelines: https://journals.plos.org/plosbiology/s/revising-your-manuscript#loc-blurb

Sincerely,

Ines

--

Ines Alvarez-Garcia, PhD

Senior Editor

PLOS Biology

Reviewers' comments:

Rev. 1:

The manuscript titled "Phase transition of a SEC14-like condensate at Arabidopsis plasma membranes regulate root growth" provides a putative mechanism that how protein polarity at the plasma membrane is determined. The authors identified an SEC-like protein (SFH8) that showed polar localization in root cells which is dependent on KISC. Further analysis showed that SFH8 functioned through a phase transition that determined PIN2 polarity which is required for root growth. It seems that the authors tried to provide many approaches to support their ideas, however, the logical flow of the story should be polished before publication. The current version of the manuscript is not friendly to either audience or reviewers. For example, they provide tons of data which are actually not explained or described in the text. We have to guess what those data referred to. Thus, I would strongly suggest the authors delete those unnecessary data and strengthen the conclusion with a simplified version of the manuscript.

1. The major finding of this story is that SFH8, as a complex component with KISC, can be phase transitioned on PM which determines PIN2 localization. Thus, the authors tried to prove that phase transition of SFH8 is required for PIN2 polarity. There is some confusion in this part. We did not see a similar phenotype in the sfh8 mutant as in the pin2 mutant. pin2 mutant showed an obvious twisted root phenotype which is not displayed in either sfh8-1 or sfh8-2(Figure S3). This majority phenotype of sfh8 is related to plant size, suggesting other mechanisms might be more important for SFH8 than regulating PIN2.

2. The authors showed that a mutation R84A of SFH8 affects its cleavage and nano clustering, which can not rescue the sfh8 mutant phenotype. This is quite interesting. I wonder whether the cleaved or non-cleaved version of SFH8 can rescue the KISC mutant phenotype.

3. Interestingly, SFH8 aligned with KISC in a filament structure. Is this aligned with the microtubule or microfilament? The authors provided some data, but it is unclear to me. This is quite important to explain how PIN2 localization is affected.

4. IDR is important for LLP in many cases. However IDR-deleted version of SFH8 can partially rescue the phenotype of the sfh8 mutant. Does this mean phase transition is not important for SFH8's function as the cleavage, since SFH8-R84A can not rescue the phenotype?

5. It is clear that PIN2 protein abundance is strongly affected in the sfh8 mutant. This can not just be ignored. All the defects can be explained simply by the low protein level of PIN2, not by mislocalization.

There are many errors in this manuscript. For example:

In line 52, what is "though although"?

In line 88, the S2A fig should be S3A fig.

Fig S1F is not explained in the manuscript. Why did they show this data?

Again, the authors have to reorganize the manuscript, delete unnecessary information, and explain the data well. I would consider it if they could provide a simple and neat version of the manuscript.

Rev. 2: Farhah Assaad – note that this reviewer has signed his review

Please use uploaded word file if possible, non-formatted text is below.

In this manuscript, Liu et al. address the role of liquid to liquid phase separation (LLPS) and liquid to solid phase transitions at the plasma membrane (PM) in Arabidopsis. This study builds on the characterization of KISC, a multiprotein complex consisting of Separase and 3 homologs of Kinesin 7 (including KIN 7.3; Moschou et al., Dev Cell 2016). The study by Liu et al. starts with interaction screens or assays (yeast two-hybrid, co-immunoprecipitation and ratiometric bimolecular fluorescence) with KIN7.3 as bait. This identified SFH8, a previously uncharacterized SEC 14 like lipid transfer protein, as a KIN7.3 interaction partner.

The authors report that

* SFH8 tethers KISC on the plasma membrane.

* In turn, KISC promotes SFH8 polarization.

* SFH8 has an Intrinsically Disordered Region (IDR) at its N-terminus, spanning 1-84aa.

* KISC cleaves SFH8 at the aa residue R84 (potential cleavage site). This releases two fragments of SFH8: (i) an N-terminal SFH8IDR and (ii) a C-terminal SFH8ΔIDR.

* Full length SFH8, an uncleavable SFH8 variant and the cleaved N-terminal SFH8IDR can be seen to form cytoplasmic puncta.

* In contrast, the cleaved C-terminal SFH8ΔIDR colocalizes with KISC and PIN2 at the plasma membrane and has a striated or filamentous appearance on confocal, TIRF and electron micrographs.

* The cleaved C-terminal SFH8ΔIDR allows the association and polar delivery of PIN2 on the PM.

* In contrast, full length SFH8 or the N-terminal SFH8IDR block PIN2 delivery.

* SFH8 and KISC mutant alleles, truncations or variants have defects in (i) root growth, (ii) gravitropism, (iii) auxin distribution and (iv) PIN2 polarity.

This is a highly novel, impressive and interesting study. There are a massive number of different experimental approaches, including targeted mutagenesis, the catGranule algorithm, FRAP rates, quantitative proximity ligation (PLA) assays and an in vitro membrane fusion assay based on cholesterol-modified DNA zippers. The manuscript makes inroads into unchartered territory.

However, the manuscript was somewhat overwhelming to read. More importantly, it is not uniformly clear that the main conclusions are supported by the dataset.

Major concerns:

1. The authors interpret SFH8IDR puncta as condensates or clusters formed via LLPS. Conversely, C-terminal SFH8ΔIDR filaments are interpreted as being formed by a liquid to solid phase transition of the SFH8 condensate, initially in the LLPS form. While this view is likely correct, it would be important to start by exploring and excluding alternative hypotheses. KISC, for example, is known to control microtubule dynamics (Moschou et al., Dev Cell 2016). The authors exclude microtubule association as underlying the striated, filamentous appearance of SFH8 on the counts of (i) colocalization and (ii) the impact of microtubule depolymerizing drugs (S4A, B Figs). This argument could be tightened and the role of actomyosin filaments excluded as well. Once alternative hypotheses are excluded, the authors could further strengthen their case by outlining the properties of LLPS and grouping the data in the manuscript that exhibit these properties. Key features of liquid-like compartments include: (a) spherical appearance, (b) fusing with each other upon touching, (c) drip in shear flows and (d) fluid internal components (Cuevas-Velazquez & Dinneny, 2018, Curr Opin Plant Biol; additional criteria such as defining saturation concentrations are described in Alberti et al., 2019, Cell).

2. In Figure 1D the authors fail to distinguish between cross walls versus cell plates. Furthermore, they use a marker (FM4-64) that does not discriminate between the middle of the cell plate versus its leading edge. Cross walls or emerging plasma membranes formed during cytokinesis are not crucial to the story line, so one could simply delete these panels.

3. The introduction and discussion are very short and the relevant literature not adequately reviewed. LLPS has been implicated in viral assembly and human disease. In plants, LLPS has been implicated in metabolism, repair, sorting and in responses to biotic and abiotic stress (doi:10.1016/j.molp.2022.05.007; doi:10.1038/s41586-019-0880-5; doi:10.1016/j.cell.2020.02.045; doi:10.1038/s41586-021-03572-6; doi:10.1016/j.cell.2020.09.010). The importance of the IDR in the plant literature could also be better outlined. To review the literature would highlight the novelty of the findings described in this manuscript.

4. The manuscript is very difficult to follow in its current shape. It would benefit from streamlining. In addition, the flow of logic should be tightened. As a suggestion, the structure could be outlined as follows: (i) identification and characterization of SFH8 (ii) cleavage of SFH8 (iii) filamentous versus punctate appearance of SFH8 full length, variants or fragments (iv) the case for LLPS or liquid to solid transitions (v) the role of LLPS in PIN polarity (including Fig. 2F, 2G), auxin distribution (Fig. 9) and plant development, including gravitropism (Figs 2G, S3A). The discussion should start with a recapitulation of the results. A graphical abstract would also help the reader.

Minor comments:

5. The figures are difficult to navigate. It would help if they could be structured in a grid-like manner and if the reader could move through them either from left to right or from top to bottom. Different annotations (as opposed to the same annotation in a different color) such as arrows, arrowheads or asterisks should be used to label structures and salient features consistently and clearly throughout the manuscript. The text font on the figures is too large, such that the labels are at times longer than the panels. Furthermore, the labels have too much information; promoters, for example, could be in the legend and the minimal relevant designator on the panel.

6. In Figure 2A the authors refer to the role of SFH8 in the "modulation of development" (an action subtitle; would a descriptive subtitle not be preferable?); here they should stay closer to their observations and describe more specifically the impact on root growth.

7. In Fig 2C the authors write "The depletion of sfh8 leads to a reduced meristem size"; it is not clear what the arrow heads point to, nor how meristem size was assessed.

8. In Fig 2D the authors write "The reduction of sfh8 meristem size is due to compromised cell divisions". However, the authors have not monitored cell expansion as a function of distance from the QC/ i.e. exit from the meristem, which also influences meristem size. The conclusion could be turned around to "compromised cell division contributes to the reduction in meristem size".

9. Fig 2G is all important as it shows an impact of KISC disruption via transient over expression of the KIN7.3 tail on gravitropism; this should be quantified as shown for other mutants in S3A Fig (circular plots showing moderate gravity perception defects of sfh8, k135, rsw4, the double rsw4 sfh8, and k135 sfh8 mutants).

10. It would simplify the main figures considerably if the authors could focus on region 3 and relegate other regions of the root to the supporting information files.

11. The legends of the figures 3B, 4A, S1B, S1C, S2A, S3A, S6A, S9B, S10A state that experiments were replicated multiple times. However, the exact number of replicates is not indicated.

---

## [Decision Letter · Decision Letter 2]

30 Jun 2023

Dear Dr Moschou,

Thank you for your patience while we considered your revised manuscript "Phase transitions of a SEC14-like condensate at Arabidopsis plasma membranes regulate root growth" for consideration as a Research Article at PLOS Biology. Please accept my apologies for the delay in providing you with our decision. Your revised study has now been evaluated by the PLOS Biology editors, the Academic Editor and the two original reviewers.

The reviews are attached below. As you will see, the reviewers agree that the manuscript has improved and, after discussing their comments with the Academic Editor, we are pleased to offer you the opportunity to address the remaining points from the reviewers in a revision that we anticipate should not take you very long. Please address all the points raised by Reviewer 1, including the removal of Fig. 10, and all Reviewer 2's comments, specially Major comments #2 and 3, and Minor comment 3.

We will then assess your revised manuscript and your response to the reviewers' comments with our Academic Editor aiming to avoid further rounds of peer-review, although might need to consult with the reviewers, depending on the nature of the revisions.

**IMPORTANT - SUBMITTING YOUR REVISION**

3. Resubmission Checklist

a) *PLOS Data Policy*

b) *Published Peer Review*

Sincerely,

Ines

--

Ines Alvarez-Garcia, PhD

Senior Editor

PLOS Biology

Reviewers' comments

Rev. 1:

The revised version of the manuscript is indeed much improved. The logic flow is much clear. A few minor concerns need to be corrected.

1. Figure 1B, for the Free YFP/RFP part, the right figure is not the merged image. Please confirm.

2. Figure 2A (right panel), is a bit confusing how the N-truncated and C-cleavage are detected in the same gel. Please clarify.

3. Figure 10 is unnecessary. It did not help the whole story to show that auxin is involved without any possible mechanism. The current story is already complicated.

4. An appropriate model might help to understand the story better.

Rev. 2:

This is a review of a revised manuscript. The authors have made some effort to address reviewer comments, but have not fully embraced our criticism, which was meant to be constructive.

The manuscript addresses the role of liquid to liquid phase separation (LLPS) and liquid to solid phase transitions at the plasma membrane (PM) in Arabidopsis. This study builds on the characterization of KISC, a multiprotein complex consisting of Separase and 3 homologs of Kinesin 7 (including KIN 7.3; Moschou et al., Dev Cell 2016). The study by Liu et al. starts with interaction screens or assays (yeast two-hybrid, co-immunoprecipitation and ratiometric bimolecular fluorescence) with KIN7.3 as bait. This identified SFH8, a previously uncharacterized SEC 14 like lipid transfer protein, as a KIN7.3 interaction partner.

The concerns listed in the review are in abbreviated form in black and my comments on the rebuttal is in blue (now indicated as ‘response’).

Major concerns:

1. The authors interpret SFH8 IDR puncta as condensates or clusters formed via LLPS. Conversely, C-terminal SFH8 IDR filaments are interpreted as being formed by a liquid to solid phase transition of the SFH8 condensate, initially in the LLPS form. While this view is likely correct, it would be important to start by exploring and excluding alternative hypotheses.

Response: The authors have done a thorough job on this first comment and it would be important to make the considerations in the rebuttal apparent in the text.

2. Response: Reference to cell plates or cross walls has been deleted, as requested.

3. The introduction and discussion are very short and the relevant literature not adequately

reviewed.

Response: The new literature includes a reference to Dhonukshe P, et al., A PLETHORA-auxin transcription module controls cell division plane rotation through MAP65 and CLASP. Cell. 2012 149(2):38396. doi: 10.1016/j.cell.2012.02.051, which has been retracted: Cell. 2013 Nov 21;155(5):1189. PMID: 22500804. It would be good to check the literature more carefully.

4. The manuscript is very difficult to follow in its current shape.

Response: The manuscript is still daunting to read and follow: it requires an inordinate amount of time and attention from the reader, which is simply not forthcoming. There is still no graphical abstract.

Minor comments:

1. The figures are difficult to navigate. It would help if they could be structured in a grid-like manner and if the reader could move through them either from left to right or from top to bottom. Different annotations (as opposed to the same annotation in a different color) such as arrows, arrowheads or asterisks should be used to label structures and salient features consistently and clearly throughout the manuscript. The text font on the figures is too large, such that the labels are at times longer than the panels. Furthermore, the labels have too much information; promoters, for example, could be in the legend and the minimal relevant designator on the panel.

Response: The figures are still not grid-like in their organization; they are still too busy and have single designations for multiple panels. The text on the panels is still conveys too much information.

2. In Figure 2A the authors refer to the role of SFH8 in the “modulation of development” (an action subtitle; would a descriptive subtitle not be preferable?); here they should stay closer to their observations and describe more specifically the impact on root growth.

Response: OK.

3. In Fig 2C the authors write “The depletion of sfh8 leads to a reduced meristem size”; it is not clear what the arrow heads point to, nor how meristem size was assessed.

Response: Here the manner in which the first elongating cell was defined is unclear.

4. In Fig 2D the authors write “The reduction of sfh8 meristem size is due to compromised cell divisions”. However, the authors have not monitored cell expansion as a function of distance from the QC/ i.e. exit from the meristem, which also influences meristem size. The conclusion could be turned around to “compromised cell division contributes to the reduction in meristem size”.

Response: OK.

5. Fig 2G is all important as it shows an impact of KISC disruption via transient over expression of the KIN7.3 tail on gravitropism; this should be quantified as shown for other mutants in S3A Fig (circular plots showing moderate gravity perception defects of sfh8, k135, rsw4, the double rsw4 sfh8, and k135 sfh8 mutants).

Response: OK.

6. It would simplify the main figures considerably if the authors could focus on region 3 and relegate other regions of the root to the supporting information files.

Response: Maybe this cannot be done everywhere, but the unnecessary regions should be deleted where possible.

7. The legends of the figures 3B, 4A, S1B, S1C, S2A, S3A, S6A, S9B, S10A state that experiments were replicated multiple times. However, the exact number of replicates is not indicated.

Response: OK.

---

## [Editor Report · Decision Letter 3]

21 Jul 2023

Dear Dr Moschou,

Thank you for your patience while we considered your revised manuscript entitled "Phase transitions of a SEC14-like condensate at Arabidopsis plasma membranes regulate root growth" for publication as a Research Article at PLOS Biology. This revised version of your manuscript has been evaluated by the PLOS Biology editors and the Academic Editor.

Based on our Academic Editor's assessment of your revision, we are likely to accept this manuscript for publication, provided you satisfactorily address the data and other policy-related requests stated below.

In addition, we would like you to consider a suggestion to improve the title:

"SEC14-like condensate phase transitions at plasma membranes regulate root growth in Arabidopsis"

We expect to receive your revised manuscript within two weeks. 

*Published Peer Review History*

*Press*

Sincerely,

Ines

--

Ines Alvarez-Garcia, PhD

Senior Editor

PLOS Biology

I am aware that you might have deposit this data already in Zenodo, however it doesn't seem to be publicly available at this stage, so I haven't been able to access it at this stage and I am not sure what is included. Here is the information we need.

The PLOS Data Policy requires that all data be made available without restriction: http://journals.plos.org/plosbiology/s/data-availability. For more information, please also see this editorial: http://dx.doi.org/10.1371/journal.pbio.1001797

Fig. 1A, E, G; Fig. 2B; Fig. 3C; Fig. 4D; Fig. 5A-C, E, G; Fig. 6A-C, E, F; Fig. 7B, C; Fig. 8E, G, I; Fig. 9B-F; Fig. S4B; Fig. S5D; Fig. S6C; Fig. S7; Fig. S8A, C; Fig. S9C, D; Fig. S10A, E, F; Fig. S11B, F, G and Fig. S12B, D, E

We require the original, uncropped and minimally adjusted images supporting all blot and gel results reported in an article's figures or Supporting Information files. We will require these files before a manuscript can be accepted so please prepare and upload them now. Please carefully read our guidelines for how to prepare and upload this data: https://journals.plos.org/plosbiology/s/figures#loc-blot-and-gel-reporting-requirements

---

## [Editor Report · Decision Letter 4]

20 Aug 2023

Dear Dr Moschou,

Thank you for the submission of your revised Research Article entitled "SEC14-like condensate phase transitions at plasma membranes regulate root growth in Arabidopsis" for publication in PLOS Biology. On behalf of my colleagues and the Academic Editor, Mark Estelle, I am delighted to let you know that we can in principle accept your manuscript for publication, provided you address any remaining formatting and reporting issues. These will be detailed in an email you should receive within 2-3 business days from our colleagues in the journal operations team; no action is required from you until then. Please note that we will not be able to formally accept your manuscript and schedule it for publication until you have completed any requested changes.

PRESS

Sincerely, 

Ines

--

Ines Alvarez-Garcia, PhD

Senior Editor

PLOS Biology
